# Calibration Attention: Instance-wise Temperature Scaling for Vision Transformers

## Abstract

Calibration is essential for deploying Vision Transformers (ViTs) in risk-sensitive settings. While post-hoc temperature scaling fits a single global scalar on a validation split, it can degrade under distribution shift, as it ignores input-dependent uncertainty. We introduce Calibration Attention (CalAttn), a lightweight plug-in head that that dynamically learns an adaptive, per-instance temperature directly from the ViT's CLS token. On CIFAR-10/100, MNIST, Tiny-ImageNet and ImageNet-1K with ViT/DeiT/Swin backbones, CalAttn reduces ECE by 2.02 pp (57.2%) pre-TS and 1.18 pp (56.6%) post-TS on average, while adding $<0.1\%$ parameters. Learned temperatures concentrate near 1.0 on in-distribution data, limiting distortion when the model is already calibrated, yet adapt on harder examples. Extensive experiments confirm robustness across datasets, while comparisons highlight CalAttn's efficiency over Dirichlet heads ($3\times$ params) and logit-temperature scaling baselines. Calibration Attention thus offers a simple, efficient strategy for producing trustworthy predictions in state-of-the-art Vision Transformers.

## 1 Introduction

A classifier is *well calibrated* when its predicted probabilities match empirical correctness: among predictions made at $80\%$ confidence, roughly $80\%$ should be correct. Calibration matters whenever confidence informs downstream actions, especially in risk-sensitive settings such as medical diagnosis Mehrtash et al. (2020), autonomous driving Feng et al. (2021), and finance Liang et al. (2024b).

**Limits of global temperature scaling.** Modern CNNs and Vision Transformers (ViTs) achieve strong accuracy yet are frequently miscalibrated Guo et al. (2017); Minderer et al. (2021); Ovadia et al. (2019). The standard remedy—post-hoc temperature scaling (TS)—fits a *single* global inverse-temperature $T^\star$ on a held-out split Guo et al. (2017). However, under distribution shift the mix of sample difficulty and uncertainty changes Ovadia et al. (2019); the optimal scaling becomes *input-dependent* (heteroscedastic). A fixed $T^\star$ that works in-distribution can then over-cool easy samples and under-cool hard ones, leading to brittle calibration despite good accuracy. Prior training-time objectives (e.g., focal variants, label smoothing, and differentiable ECE surrogates) help but still impose static hyperparameters and a single global scale Mukhoti et al. (2020a); Müller et al. (2019); Tao et al. (2023); Liang et al. (2024a).

**A representation-conditioned signal for uncertainty.** Transformers expose a compact summary via the final `[CLS]` embedding. We observe that its $\ell_2$ norm, $\|\mathbf{z}_{\text{CLS}}\|_2$, correlates with confidence (e.g., Pearson $r \approx 0.45$; Fig. 2), suggesting it encodes sample difficulty. This motivates *instance-wise* calibration that conditions the temperature on $\mathbf{z}_{\text{CLS}}$ rather than holding it fixed.

**Calibration Attention (CalAttn).** We introduce a lightweight two-layer MLP ($< 0.1\%$ parameters) that maps $\mathbf{z}_{\text{CLS}}$ to a positive scale $s(\mathbf{z})$ and uses it to re-scale logits during *both* training and inference. CalAttn is trained end-to-end with the backbone using cross-entropy plus a small Brier penalty, thereby (i) adapting confidence *per sample* and (ii) delivering *calibration gradients* to the representation. When the data distribution shifts, the representation changes and $s(\mathbf{z})$ tracks it, yielding instance-adaptive calibration where global optimal $T^\star$ is brittle.

**Contributions.** (i) We propose **CalAttn**, a representation-conditioned, drop-in calibration head for ViT-style classifiers that integrates seamlessly with standard training. (ii) We provide quantitative evidence that `[CLS]` carries usable uncertainty cues and exploit them to learn $s(\mathbf{z})$. (iii) Across

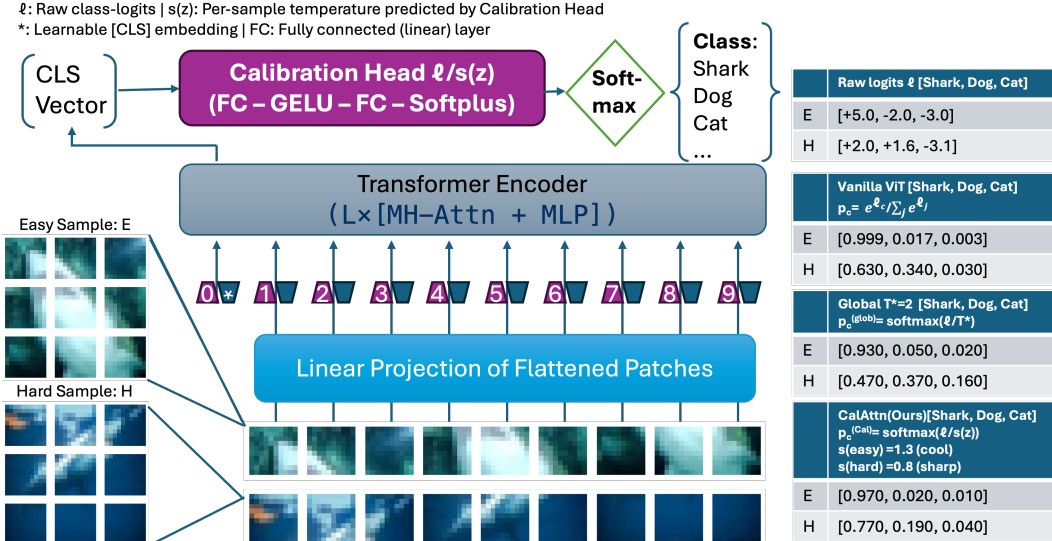

Figure 1: **ViT +CalibrationAttention (CalAttn).** A standard Vision Transformer pipeline is augmented with a lightweight Calibration Head (FC + GELU + FC + Softplus) that reads the final `CLS` embedding and predicts a strictly–positive, *per-sample* temperature $s(\mathbf{z})$. The class logits produced by the usual classifier head are divided by this scale before the Soft-max, enabling image-adaptive cooling or sharpening of the output distribution. The main backbone mirrors the architecture of Dosovitskiy et al. (2021); the tables on the right are illustrated for a 3-class task.

CIFAR-10/100, Tiny-ImageNet, MNIST, and ImageNet 1K, CalAttn improves ECE and smooth-ECE while preserving accuracy; it further composes with post-hoc TS and SATS, yielding additional gains (Sec. 3).

## 2 METHOD

To address the above limitations, our aim is to replace the *single, global* temperature used in post-hoc scaling with an *instance-wise*, image-adaptive temperature that is learned jointly with model parameters. Fig. 1 presents an overview of our approach, and full notation is provided in Appendix A.

### 2.1 PRELIMINARIES

**Vision-Transformer logits.** Given an RGB image $\mathbf{x} \in \mathbb{R}^{3 \times H \times W}$, a ViT prepends a learned `[CLS]` token and produces, after $L$ encoder blocks, a global embedding $\mathbf{z}_{\text{CLS}} \in \mathbb{R}^d$. A linear head $\boldsymbol{\ell} = \mathbf{W}_{\text{cls}}\mathbf{z}_{\text{CLS}} + b_{\text{cls}}$ yields class-logits $\boldsymbol{\ell} \in \mathbb{R}^C$, which are turned into probabilities by a soft-max with temperature $T$,

$$\sigma_i(\boldsymbol{\ell}, T) = \frac{\exp(\ell_i/T)}{\sum_{j=1}^C \exp(\ell_j/T)}. \tag{1}$$

**Limitations of global scaling.** Temperature scaling (Guo et al., 2017) fits a *single* scalar $T^\star$ on a held-out set and deploys $\hat{y}_i = \sigma_i(\boldsymbol{\ell}, T^\star)$ at test time. Because $T^\star$ is constant, it cannot reconcile the heteroscedastic [1] confidence of individual samples, and the backbone never sees calibration gradients.

### 2.2 CALIBRATION ATTENTION (CALATTN)

**A per-sample temperature head.** After the Transformer encoder, the first token yields a global classification embedding $\mathbf{z} \in \mathbb{R}^d$, commonly known as the CLS token. *Calibration Attention*

---

[1]Easy images lead to large $\|\mathbf{z}_{\text{CLS}}\|_2$ and over-confidence; difficult images do the opposite.

*(CalAttn)* maps this embedding through a lightweight two-layer MLP to predict an adaptive, per-sample temperature:

$$s(\mathbf{z}) = \text{softplus}\left(\mathbf{w}_2^\top \text{GELU}(\mathbf{W}_1\mathbf{z}) + b_2\right) + \varepsilon \tag{2}$$

where $\mathbf{W}_1 \in \mathbb{R}^{h \times d}$, $\mathbf{w}_2 \in \mathbb{R}^h$, $b_2 \in \mathbb{R}$, and $\varepsilon = 10^{-6}$. Parameters are initialized as $\mathbf{w}_2 \leftarrow \mathbf{0}$ and $b_2 \leftarrow \ln(e^1 - 1)$, ensuring $\text{softplus}(b_2) \approx 1$. Thus, *CalAttn initially matches the baseline model and deviates only when driven by calibration improvements.*

The final calibrated probability distribution is computed as: $\widehat{\boldsymbol{y}} = \text{softmax}\left(\frac{\boldsymbol{\ell}}{s(\mathbf{z}_{\text{CLS}})}\right)$, where scaling by the adaptive scalar $s(\mathbf{z})$ preserves the original class ranking, dynamically cooling overly confident predictions ($s > 1$) or sharpening under-confident ones ($s < 1$).

**MLP Design.** The choice of a small two-layer MLP for CalAttn is guided by several key insights:

1. **Implicit Difficulty Signal in z.** Empirically, the magnitude and geometry of the CLS embedding encode information about sample difficulty. CalAttn directly leverages this intrinsic representation to predict adaptive calibration scales.

2. **Expressivity with Minimal Cost.** A single hidden-layer MLP with GELU activation universally approximates scalar functions. With a modest width $h = 128$, this module adds only $hd + d + h + 1$ parameters, equivalent to less than $0.1\%$ of the parameters in typical Transformers (e.g., DeiT-Small), ensuring negligible computational overhead.

3. **Smooth and Positive Scaling via Softplus.** Softplus ensures strictly positive scaling ($s(\mathbf{z}) > 0$), provides smooth and stable gradients, and avoids numerical instabilities common with non-smooth alternatives like ReLU (Glorot et al., 2011). The small offset $\varepsilon$ further guarantees it.

4. **Calibration-Aligned Gradient Dynamics.** For a combined cross-entropy and Brier loss (Eq. 4), the gradient w.r.t. the scaling parameter is [2]:

$$\frac{\partial \mathcal{L}}{\partial s} = (\hat{y}_{\hat{c}} - \mathbf{1}[y = \hat{c}])\left(\ell_{\hat{c}} - \sum_j \hat{y}_j \ell_j\right) s^{-1}. \tag{3}$$

   This gradient term is positive when predictions are incorrect yet overly confident (thus increasing $s$), and negative when correct but underconfident (decreasing $s$), directly aligning confidence with accuracy to reduce calibration error.

**Heteroscedastic-Noise Perspective.** CalAttn can also be justified statistically. Assume raw logits scale inversely with a latent, per-image noise level $\sigma(x)$, i.e., $\boldsymbol{\ell} \propto 1/\sigma(x)$. The Bayes-optimal softmax would thus use temperature scaling $T(x) \propto \sigma(x)$.

CalAttn achieves this *ideal scenario*[3] end-to-end by optimizing a differentiable calibration-aware objective:

$$\mathcal{L} = \mathcal{L}_{\text{CE}} + \lambda \left\|\widehat{\boldsymbol{y}} - \mathbf{e}_y\right\|_2^2, \tag{4}$$

where $\mathbf{e}_y$ is the one-hot ground-truth vector and $\lambda = 0.1$. Also, the Brier term serves as a differentiable proxy for Expected Calibration Error (ECE) (Liang et al., 2024a; Blasiok et al., 2023). As the only scalar that can modify confidence while preserving class ranks, the learned temperature $s(\mathbf{z}_{\text{CLS}})$ naturally converges toward the optimal temperature (Proof in Appendix E):

$$\nabla_s \mathcal{L} = 0 \quad \Longrightarrow \quad s(\mathbf{z}_{\text{CLS}}) \approx T(x)_{\text{optimal}}. \tag{5}$$

Thus, CalAttn effectively replaces traditional two-stage post-hoc temperature searches with a one-stage, fully differentiable calibration solution as depicted in Table 4.

---

[2] where $\hat{c} = \arg\max_i \ell_i$ is the predicted class, $\hat{y}_i = \exp(\ell_i/s)/\sum_j \exp(\ell_j/s)$ the soft-max probability for class $i$, $\mathbf{1}[y = \hat{c}] = 1$ if the true label $y$ equals $\hat{c}$ and 0 otherwise, and $\ell_j$ denotes the $j$-th logit before scaling.

[3] "ideal" mean the Bayes-optimal, instance-wise temperature $T^\star(x) \propto \sigma(x)$, where $\sigma(x)$ is the latent per-sample noise scale: this unique scalar exactly aligns a model's predicted confidence with its true accuracy.

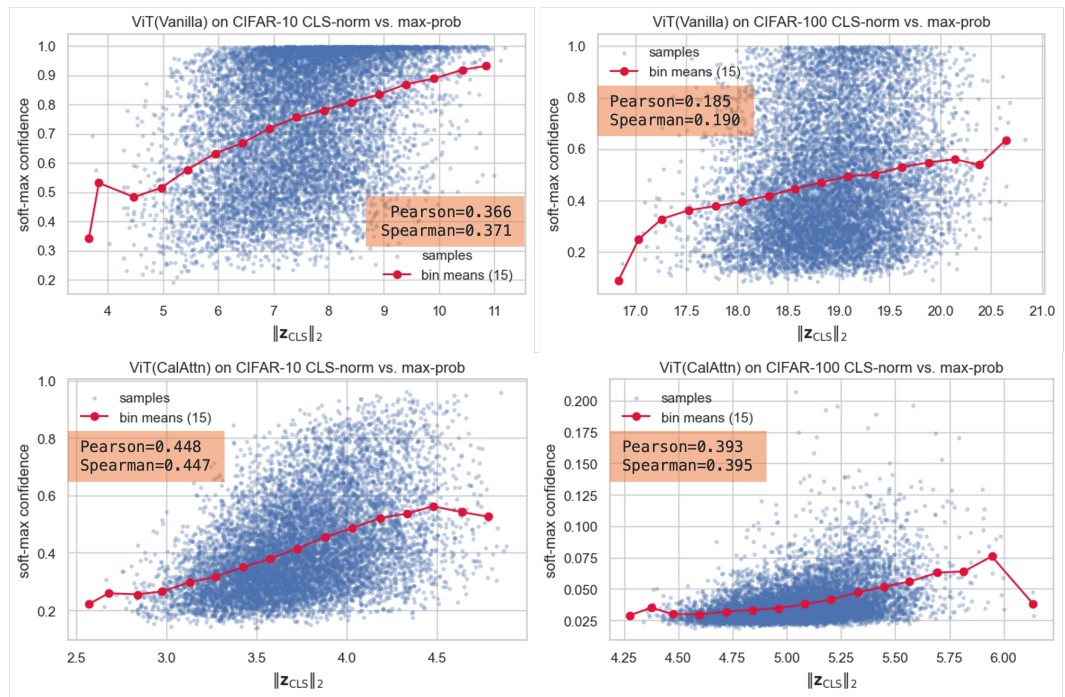

Figure 2: **CLS-norm versus soft-max confidence** on CIFAR-10/100 with ViT-224. Blue: individual images; red: mean in 15 equal-width bins. The positive correlation shows that $\|\mathbf{z}_{\mathrm{CLS}}\|_2$ diagnoses sample difficulty, which CalAttn converts into an adaptive temperature.

## 2.3 CLS TOKEN IS A GOOD "THERMOMETER"

**Global summary token.** In ViT–style architectures a *class* token is prepended to the patch sequence and attends to *all* patches at every layer. After $L$ encoder blocks the first-token embedding $\mathbf{z}_{\mathrm{CLS}} \in \mathbb{R}^d$ is a compact, information-rich summary: a linear head produces the logits $\boldsymbol{\ell} = \mathbf{W}\mathbf{z}_{\mathrm{CLS}}$. Empirically, the norm and direction of $\mathbf{z}_{\mathrm{CLS}}$ correlate with (i) image difficulty, (ii) inter-class margin and (iii) distribution shift (Naseer et al., 2021; Zhou et al., 2022; Minderer et al., 2021).

**Logit scale and confidence.** For a linear classifier the *magnitude* of the pooled token is a proxy for the scale of the logits:

$$\big\|\mathbf{z}_{\mathrm{CLS}}\big\|_2 \xrightarrow{\ \mathbf{W}\cdot\ } \text{ larger logits } \xrightarrow{\ \sigma(\cdot)\ } \text{ higher soft-max confidence.}$$

Empirically (Figure 2 plots the L2 norm $\|\mathbf{z}_{\mathrm{CLS}}\|_2$ against soft-max confidence for CIFAR-10/100.) we observe a clear, monotonic trend: Easy images with large $\|\mathbf{z}_{\mathrm{CLS}}\|_2$ tend to be over-confident; while hard images with small $\|\mathbf{z}_{\mathrm{CLS}}\|_2$ tend to be under-confident. A single post-hoc temperature $T^\star$ inevitably calibrates only a narrow slice of this spectrum, leaving both tails mis-calibrated. The moderate monotonic trend (Pearson $r=0.45$, Spearman $\rho=0.45$) confirms that ViTs already encode sample difficulty; CalAttn exploits this signal through Eq. 2.

## 2.4 TEMPERATURE INFERENCE

At test time we run one forward pass: compute $s(\mathbf{z})$, divide logits, apply soft-max. No extra tuning or ensemble passes are required (See Algorithm 1). CalAttn's simplicity, tiny footprint and substantial drops in ECE (Table 1) make it a practical *drop-in module* for trustworthy Vision Transformers.

---

**Algorithm 1** Per-sample Temperature Scaling with **Calibration Attention**

---

**Require:** CLS token $\mathbf{z}_{\mathrm{cls}} \in \mathbb{R}^d$, logits $\boldsymbol{\ell} \in \mathbb{R}^C$, weights $W_1 \in \mathbb{R}^{h \times d}$, $W_2 \in \mathbb{R}^{1 \times h}$, bias $b_2 = \ln(e^1 - 1)$
    ▷ Softplus$(b_2) \approx 1$
**Ensure:** Calibrated class-probabilities $\hat{\mathbf{y}} \in \mathbb{R}^C$
1: $\mathbf{h} \leftarrow \mathrm{GELU}(W_1 \mathbf{z}_{\mathrm{cls}})$                                    ▷ first FC: $d \rightarrow h$
2: $s \leftarrow \mathrm{Softplus}(W_2 \mathbf{h} + b_2) + \varepsilon$                         ▷ $s > 0$, $\varepsilon = 10^{-6}$
3: $\boldsymbol{\ell}^{\mathrm{cal}} \leftarrow \boldsymbol{\ell}/s$                        ▷ element-wise cooling/sharpening of logits
4: $\hat{\mathbf{y}} \leftarrow \mathrm{softmax}(\boldsymbol{\ell}^{\mathrm{cal}})$
5: **Return** $\hat{\mathbf{y}}$

---

## 3 EXPERIMENTS

### 3.1 EXPERIMENTAL SET-UP

Table 1: ↓ ECE (%) for methods both before and after applying temperature scaling (Bin = 15) [3].

| Dataset | Model | Weight Decay Guo et al. (2017) | | Brier Loss Brier (1950b) | | MMCE Kumar et al. (2018) | | Label Smooth Szegedy et al. (2016) | | Focal Loss - 53 Mukhoti et al. (2020a) | | Dual Focal Tao et al. (2023) | | CE + $\lambda$ BS (Ours*, $\lambda = 0.1$) | |
|---|---|---|---|---|---|---|---|---|---|---|---|---|---|---|---|
| | | Pre T | Post T | Pre T | Post T | Pre T | Post T | Pre T | Post T | Pre T | Post T | Pre T | Post T | Pre T | Post T |
| | ResNet-50 | 18.02 | 2.60(2.2) | 5.47 | 3.71(1.1) | 15.05 | 3.41(1.9) | 6.38 | 5.12(1.1) | 5.54 | 2.28(1.1) | 8.79 | 2.31(1.3) | 14.41 | 2.35(1.60) |
| | ResNet-110 | 19.29 | 4.75(2.3) | 6.72 | 3.59(1.2) | 18.84 | 4.52(2.3) | 9.53 | 5.20(1.3) | 11.02 | 3.88(1.3) | 11.65 | 3.75(1.3) | 19.36 | 3.38(1.80) |
| | Wide-ResNet-26-10 | 15.17 | 2.82(2.1) | 4.07 | 3.03(1.1) | 13.57 | 3.89(2.0) | 3.29 | 3.29(1.0) | 2.42 | 2.16(1.1) | 5.30 | 2.27(1.2) | 6.63 | 5.24(1.20) |
| | DenseNet-121 | 19.07 | 3.42(2.2) | 4.28 | 2.22(1.1) | 17.37 | 3.22(2.0) | 8.63 | 4.82(1.1) | 3.40 | **1.55(1.1)** | 6.69 | 1.65(1.2) | 16.10 | 2.95(1.60) |
| CIFAR-100 | ViT$_{224}$ | 13.11 | 3.30(1.30) | **2.99** | 2.99(1.00) | 11.52 | 2.34(1.30) | 3.83 | **2.24(1.10)** | 5.33 | 3.46(1.10) | 3.42 | 3.42(1.00) | 6.89 | 3.49(1.50) |
| | ViT$_{224}$+CalAttn(Ours) | 8.25 | 1.85(1.20) | 1.98 | 2.15(1.10) | 9.04 | 2.22(1.20) | **2.09** | 2.09(1.00) | 2.61 | 2.61(1.00) | 3.72 | 1.92(1.10) | 4.09 | **1.49(1.10)** |
| | DeiT$_{small}$ | 7.46 | 3.54(1.10) | 2.17 | 2.17(1.00) | 7.88 | 2.55(1.20) | **1.75** | **1.75(1.00)** | 3.27 | 3.27(1.00) | 1.93 | 1.93(1.00) | 5.46 | 3.86(1.40) |
| | DeiT$_{small}$+CalAttn(Ours) | 6.87 | 1.48(1.20) | 1.98 | 2.19(1.10) | 6.51 | 1.86(1.20) | 1.55 | 1.55(1.00) | 3.15 | 3.15(1.00) | 3.30 | 2.19(1.10) | 2.88 | **0.86(1.10)** |
| | Swin$_{small}$ | 3.51 | 2.46(0.90) | 2.54 | 2.54(1.00) | 3.53 | 2.78(0.90) | 9.00 | **2.41(0.80)** | 10.64 | 2.51(0.80) | 8.18 | 3.15(0.80) | 4.59 | 2.51(1.20) |
| | Swin$_{small}$+CalAttn(Ours) | 1.93 | 1.93(1.00) | 3.03 | 2.02(1.10) | 1.92 | 1.92(1.00) | 6.59 | 2.57(0.90) | 6.15 | 3.17(0.90) | 3.97 | 3.21(0.90) | **1.51** | 1.51(1.00) |
| | ResNet-50 | 4.23 | 1.37(2.5) | 1.83 | 0.96(1.1) | 4.67 | 1.11(2.6) | 4.23 | 1.87(0.9) | 1.46 | 1.46(1.0) | 1.32 | 1.32(1.0) | 5.71 | 1.53(1.90) |
| | ResNet-110 | 4.86 | 1.94(2.6) | 2.50 | 1.36(1.2) | 5.22 | 1.24(2.8) | 3.74 | 1.22(0.9) | 1.67 | 1.10(1.1) | 1.48 | 1.27(1.1) | 6.23 | 2.29(1.90) |
| | Wide-ResNet-26-10 | 3.24 | **0.86(2.2)** | 1.07 | 1.07(1.0) | 3.60 | 1.26(2.2) | 4.66 | 1.27(0.8) | 1.83 | 1.17(0.9) | 3.35 | 0.94(0.8) | 3.49 | 1.15(1.50) |
| | DenseNet-121 | 4.70 | 1.54(2.4) | 1.24 | 1.24(1.0) | 4.97 | 1.71(2.4) | 4.05 | 1.01(0.9) | 1.73 | 1.22(0.9) | **0.70** | 1.31(0.9) | 5.16 | 1.83(1.60) |
| CIFAR-10 | ViT$_{224}$ | 7.39 | 1.68(1.30) | 1.87 | 1.87(1.00) | 3.47 | **1.06(1.10)** | 1.79 | 1.79(1.00) | 8.70 | 2.11(0.70) | 3.71 | 2.15(0.90) | 3.98 | 3.41(1.40) |
| | ViT$_{224}$+CalAttn(Ours) | 4.76 | 1.38(1.20) | 2.62 | 1.77(1.10) | 4.36 | 1.64(1.20) | 2.58 | 1.55(0.90) | 10.04 | 2.06(0.70) | 4.57 | 2.86(0.90) | 1.56 | **1.21(1.10)** |
| | DeiT$_{small}$ | 4.42 | **1.52(1.20)** | 2.02 | 2.02(1.00) | 6.01 | 1.78(1.20) | 1.84 | 1.84(1.00) | 10.65 | 2.79(0.70) | 5.11 | 2.79(0.80) | 4.25 | 2.95(1.30) |
| | DeiT$_{small}$+CalAttn(Ours) | 4.10 | 1.67(1.10) | 1.98 | 1.32(1.10) | 3.57 | 1.32(1.10) | 2.41 | 2.41(1.00) | 9.91 | 1.94(0.70) | 3.87 | 2.84(0.90) | 1.88 | **1.22(1.10)** |
| | Swin$_{small}$ | 1.25 | 1.91(0.90) | 1.12 | **1.12(1.00)** | 1.73 | 2.31(0.90) | 6.71 | 1.30(0.80) | 15.70 | 2.32(0.60) | 8.54 | 2.40(0.70) | 2.01 | 1.47(1.20) |
| | Swin$_{small}$+CalAttn(Ours) | 1.74 | 1.74(1.00) | 1.08 | 1.08(1.00) | 1.00 | 1.00(1.00) | 6.09 | 1.45(0.80) | 14.97 | 1.66(0.60) | 7.50 | 2.32(0.80) | 0.94 | **0.94(1.00)** |
| | ViT$_{224}$ | **2.91** | 1.15(0.90) | 48.83 | 5.61(8.70) | 19.98 | 5.18(2.10) | 12.77 | 8.07(1.50) | 24.07 | 2.07(1.40) | 18.12 | 4.85(2.30) | 5.34 | 1.08(0.80) |
| | ViT$_{224}$+CalAttn(Ours) | 2.73 | 0.99(0.90) | 3.13 | 0.90(0.90) | 3.89 | 1.09(0.80) | 8.29 | 0.99(0.70) | 7.10 | 1.30(0.40) | 18.37 | 1.56(0.50) | **0.74** | **0.44(0.90)** |
| MNIST | DeiT$_{small}$ | 2.87 | 1.15(0.90) | 28.37 | 2.73(3.20) | 4.66 | 1.16(0.80) | 8.70 | 1.47(0.70) | 21.72 | 2.43(0.50) | 15.90 | 1.55(0.70) | 4.48 | 1.04(0.80) |
| | DeiT$_{small}$+CalAttn(Ours) | 1.30 | 1.30(1.00) | 1.15 | 0.46(0.90) | 2.11 | 0.92(0.90) | 6.20 | 1.00(0.60) | 18.28 | 1.54(0.50) | 8.96 | 2.28(0.60) | **0.91** | 0.91(1.00) |
| | Swin$_{small}$ | 3.64 | 3.21(1.20) | 1.32 | 0.67(0.90) | 1.26 | 0.67(0.90) | 8.09 | 0.56(0.60) | 7.09 | 1.28(0.50) | 5.04 | 0.47(0.60) | 0.46 | 0.46(1.00) |
| | Swin$_{small}$+CalAttn(Ours) | **0.66** | 0.56(1.10) | 0.85 | 0.45(0.90) | 1.16 | 0.51(0.7) | 6.89 | 0.46(0.6) | 11.75 | 0.84(0.50) | 8.78 | 0.49(0.50) | 0.85 | **0.42(0.90)** |
| | ViT$_{224}$ | 22.11 | 1.93(1.90) | 4.44 | 1.89(1.10) | 19.84 | 1.81(1.80) | **2.25** | 1.51(1.10) | 3.28 | 1.93(1.00) | 2.45 | 2.44(1.10) | 1.75 | 1.66(1.10) |
| | ViT$_{224}$+CalAttn(Ours) | 4.41 | 3.03(1.10) | 9.18 | 1.78(1.30) | 4.97 | 3.07(1.20) | 2.22 | 1.48(1.10) | 1.68 | 1.14(1.30) | 13.57 | 1.34(1.40) | 1.01 | **0.78(1.10)** |
| Tiny-ImageNet | DeiT$_{small}$ | 13.04 | 2.13(1.10) | 2.70 | 1.79(1.10) | 13.66 | 1.25(1.10) | **1.64** | **1.64(1.00)** | 10.28 | 2.15(1.50) | 9.37 | 1.79(1.30) | 1.69 | 1.55(1.10) |
| | DeiT$_{small}$+CalAttn(Ours) | 2.37 | 1.38(1.40) | 1.09 | 0.19(1.20) | 2.83 | 1.26(1.50) | 2.36 | 1.20(1.10) | 2.15 | 1.11(1.30) | 1.79 | 0.87(1.30) | **0.95** | 0.83(1.20) |
| | Swin$_{small}$ | 3.03 | 2.76(0.90) | **0.81** | 0.25(0.90) | 1.94 | 1.94(1.00) | 4.92 | 2.41(0.90) | 4.47 | 2.46(0.90) | 4.90 | 2.57(0.90) | 1.52 | 1.52(1.00) |
| | Swin$_{small}$+CalAttn(Ours) | 2.92 | 1.22(1.10) | 0.67 | 0.64(1.10) | 1.41 | 1.41(1.00) | 2.67 | 2.67(1.00) | 2.32 | 2.32(1.00) | 2.29 | 2.29(1.00) | 0.82 | **0.24(0.90)** |

**Datasets.** CIFAR-10 and CIFAR-100 (50k/10k images, $32^2$ px) serve as high–variance, limited-data settings where calibration is notoriously difficult. To widen the difficulty spectrum we also evaluate on 200-class subset of ImageNet(Tiny) and MNIST. **Architectures.** CNNs: ResNet-50/110, Wide-ResNet-26-10, and DenseNet-121 ($\approx$ 4M–23M parameters). Transformers: ViT$_{224}$(Dosovitskiy et al., 2021), DeiT-Small(Touvron et al., 2021), and Swin-Small(Liu et al., 2021) and our method implements identical backbones with a *single* CalAttn head. **Training.** Unless stated otherwise, we train all models for 350 epochs using SGD (momentum 0.9, weight decay $5 \times 10^{-4}$). Learning rate set to 0.1 for the first 150 epochs, 0.01 for the following 100 epochs, and 0.001 for the remaining epochs. We retain the original data-augmentation and optimisation hyper-parameters of each backbone. CalAttn uses $\lambda = 0.1$ in Eq. equation 4 for *all* experiments. No extra tricks or sweeps are introduced. **Baselines.** We re-implement seven popular calibration losses: Weight Decay (WD) (Guo et al., 2017), Brier Score (BS) (Brier, 1950b), MMCE (Kumar et al., 2018), Label Smoothing (LS) (Szegedy et al., 2016), FocalLoss-53 (Mukhoti et al., 2020a), Dual Focal Loss (DFL) (Tao et al., 2023), and Ours CE+$\lambda$BS baseline, using Brier score Brier (1950b) as a strictly proper loss, which is an *ablation* that

---

[3]We follow the experiment setup of Mukhoti et al. (2020a), utilizing their public code Mukhoti et al. (2020b). We performed cross-validation to determine the optimal $\gamma$ for each experiment. Optimal single scalar temperature scaling $\boldsymbol{\ell}/T$ by minimizing validation ECE Mukhoti et al. (2020a) (reported in parentheses in all "Post-T" aligning prior work) close to 1.0 indicates well-calibrated predictions requiring minimal adjustment.

jointly optimizes cross-entropy and Brier. Following the standard temperature-scaling framework of Guo et al. (2017), we calibrate each trained model with a single scalar $T$ estimated on a held-out validation split (we use 5% of the training data unless stated otherwise). Consistent with Mukhoti et al. (2020a), we select $T^\star$ by grid search $T \in \{0.1, 0.2, \dots, 10.0\}$ to *minimise* validation ECE (15 bins); prior work reports this procedure yields stronger baselines than NLL-based fitting, and we follow that choice for our main tables. We follow the random seed setting from prior public SOTA works (Wang et al., 2021; Kumar et al., 2018; Mukhoti et al., 2020a; Szegedy et al., 2016; Tao et al., 2023).

Table 2: ↓ smCE (%) for methods both before and after applying temperature scaling (Bin = 15) [4].

| Dataset | Model | Weight Decay Guo et al. (2017) | | Brier Loss Brier (1950b) | | MMCE Kumar et al. (2018) | | Label Smooth Szegedy et al. (2016) | | Focal Loss - 53 Mukhoti et al. (2020a) | | Dual Focal Tao et al. (2023) | | CE + λ BS (Ours*, λ = 0.1) | |
|---|---|---|---|---|---|---|---|---|---|---|---|---|---|---|---|
| | | Pre T | Post T | Pre T | Post T | Pre T | Post T | Pre T | Post T | Pre T | Post T | Pre T | Post T | Pre T | Post T |
| CIFAR-100 | ResNet-50 | 14.95 | 2.63(2.2) | 5.34 | 3.53(1.1) | 13.40 | 3.12(1.9) | 6.31 | 3.43(1.1) | 5.56 | 2.37(1.1) | 8.82 | 2.24(1.3) | 13.99 | 2.31(1.60) |
| | ResNet-110 | 15.05 | 3.84(2.3) | 6.53 | 3.51(1.2) | 14.86 | 3.51(2.3) | 9.55 | 4.36(1.3) | 10.98 | 3.60(1.3) | 11.69 | 3.31(1.3) | 17.84 | 3.16(1.80) |
| | Wide-ResNet-26-10 | 12.97 | 2.85(2.1) | 4.06 | 2.78(1.1) | 12.27 | 3.85(2.0) | 3.04 | 3.04(1.0) | **2.34** | 2.20(1.1) | 6.01 | 2.41(0.9) | 6.45 | 5.12(1.20) |
| | DenseNet-121 | 15.42 | 2.61(2.2) | 3.86 | 1.93(1.1) | 15.96 | 2.95(2.0) | 3.85 | 3.68(1.1) | 3.04 | **1.50(1.1)** | 4.07 | 1.72(0.9) | 15.75 | 2.83(1.60) |
| | ViT₂₂₄ | 13.07 | 3.32(1.30) | **2.62** | 2.25(1.10) | 11.51 | 2.25(1.30) | 3.77 | **2.17(1.10)** | 5.13 | 3.20(1.10) | 3.43 | 3.43(1.00) | 6.27 | 2.25(1.50) |
| | **ViT₂₂₄+CalAttn(Ours)** | 8.24 | 1.92(1.20) | **1.93** | 2.30(1.10) | 9.04 | 2.16(1.20) | 1.97 | 1.97(1.00) | 2.64 | 2.64(1.00) | 3.66 | 1.96(1.10) | **4.00** | **1.49(1.10)** |
| | DeiT₍small₎ | 7.45 | 3.48(1.10) | 2.13 | 2.13(1.00) | 7.88 | 2.34(1.20) | **1.72** | **1.72(1.00)** | 3.01 | 3.01(1.00) | 1.91 | 1.91(1.00) | 5.13 | 2.06(1.40) |
| | **DeiT₍small₎+CalAttn(Ours)** | 6.87 | 1.59(1.20) | 2.13 | 2.05(1.10) | 6.51 | 1.28(1.20) | **1.61** | 1.61(1.00) | 3.08 | 3.08(1.00) | 3.05 | 2.16(1.10) | 4.37 | **1.17(1.10)** |
| | Swin₍small₎ | 3.44 | 2.44(0.90) | 2.55 | 2.55(1.10) | 3.37 | 2.66(0.90) | **2.35** | 2.35(0.80) | 10.55 | 2.42(0.80) | 8.17 | 3.04(0.80) | 3.23 | 2.45(1.20) |
| | **Swin₍small₎+CalAttn(Ours)** | 1.99 | 1.99(1.00) | 3.02 | 1.92(1.10) | 1.76 | 1.76(1.00) | 6.48 | 2.56(0.90) | 5.99 | 2.99(0.90) | 3.76 | 2.92(0.90) | **1.64** | **1.64(1.00)** |
| CIFAR-10 | ResNet-50 | 3.27 | 1.48(2.5) | 1.83 | 1.27(1.1) | 3.41 | 1.47(2.6) | 3.31 | 1.92(0.9) | 1.45 | 1.45(1.0) | **1.36** | 1.36(1.0) | 5.18 | 1.67(1.90) |
| | ResNet-110 | 3.43 | 1.88(2.6) | 2.49 | 1.63(1.2) | 3.38 | 1.55(2.8) | 2.87 | 1.52(0.9) | 1.69 | 1.39(1.1) | 1.48 | 1.35(1.1) | 5.28 | 2.25(1.90) |
| | Wide-ResNet-26-10 | 2.70 | 1.25(2.2) | 1.55 | 1.55(1.0) | 3.03 | 1.46(2.2) | 3.75 | 1.47(0.8) | 1.84 | 1.46(0.9) | 3.31 | **1.20(0.8)** | 3.41 | 1.64(1.50) |
| | DenseNet-121 | 15.42 | 2.61(2.4) | 1.89 | 1.93(1.0) | 15.96 | 2.95(2.4) | 2.84 | 1.63(0.9) | 3.04 | 1.50(0.9) | 3.66 | 1.72(0.9) | 5.17 | 2.24(1.60) |
| | ViT₂₂₄ | 7.40 | 1.46(1.30) | 1.89 | 1.89(1.00) | 3.47 | **1.37(1.10)** | 1.66 | 1.66(1.00) | 8.53 | 1.97(0.70) | 3.66 | 2.12(0.90) | 4.26 | 3.53(1.40) |
| | **ViT₂₂₄+CalAttn(Ours)** | 4.75 | 1.54(1.20) | 2.56 | 1.86(1.10) | 4.36 | 1.70(1.20) | 2.41 | 1.51(0.90) | 9.95 | 2.06(0.70) | 4.35 | 2.78(0.90) | **1.57** | **1.18(1.10)** |
| | DeiT₍small₎ | 4.41 | **1.56(1.20)** | 2.03 | 2.03(1.00) | 6.01 | 1.65(1.20) | **1.64** | 1.64(1.00) | 10.49 | 2.87(0.70) | 5.00 | 2.79(0.80) | 3.85 | 1.96(1.30) |
| | **DeiT₍small₎+CalAttn(Ours)** | 4.07 | 1.76(1.10) | 2.09 | 1.76(1.10) | 3.39 | 1.41(1.10) | 2.15 | 2.15(1.00) | 9.90 | 1.79(0.70) | 3.77 | 2.74(0.90) | **1.89** | **1.36(1.10)** |
| | Swin₍small₎ | **1.43** | 1.93(0.90) | 1.46 | 1.46(1.00) | 1.78 | 2.31(0.90) | 6.68 | **1.36(0.80)** | 15.63 | 2.34(0.60) | 8.40 | 2.41(0.70) | 3.11 | 2.04(1.20) |
| | **Swin₍small₎+CalAttn(Ours)** | 1.63 | 1.63(1.00) | 1.19 | 1.19(1.00) | 1.23 | 1.23(1.00) | 5.94 | 1.55(0.80) | 14.94 | 1.61(0.60) | 7.08 | 2.32(0.80) | **1.11** | **1.11(1.00)** |
| MNIST | ViT₂₂₄ | 2.88 | 1.13(0.90) | 39.17 | 3.72(8.70) | 19.77 | 4.89(2.10) | 12.74 | 6.61(1.50) | 7.03 | 2.15(1.40) | 18.28 | 4.52(2.30) | 5.23 | **1.26(0.80)** |
| | **ViT₂₂₄+CalAttn(Ours)** | 2.76 | 1.13(0.90) | 3.12 | 1.04(0.90) | 3.89 | 1.41(0.80) | 8.30 | 1.31(0.70) | 23.30 | 1.25(0.40) | 17.99 | 1.45(0.50) | **0.94** | **0.68(0.90)** |
| | DeiT₍small₎ | 2.87 | 1.29(0.90) | 26.66 | 2.53(3.20) | 4.56 | 1.17(0.80) | 21.32 | 2.39(0.50) | 15.89 | **1.65(0.70)** | 4.45 | 1.30(0.80) | 4.45 | 1.30(0.80) |
| | **DeiT₍small₎+CalAttn(Ours)** | 1.23 | 1.23(1.00) | 1.17 | 0.68(0.90) | 2.15 | 1.13(0.90) | 6.21 | 1.40(0.80) | 18.17 | 1.56(0.50) | 8.96 | 2.30(0.60) | **1.14** | **1.14(1.00)** |
| | Swin₍small₎ | 3.57 | 3.20(1.10) | 1.28 | 0.71(0.90) | 1.35 | 0.79(0.90) | 8.12 | 0.79(0.70) | 11.77 | 0.96(0.50) | 5.06 | 0.58(0.60) | **0.65** | 0.65(1.00) |
| | **Swin₍small₎+CalAttn(Ours)** | **0.53** | 0.66(1.10) | 0.90 | 0.59(0.90) | 1.22 | 0.68(0.90) | 7.89 | 0.67(0.70) | 7.11 | 0.58(0.50) | 8.81 | 0.66(0.50) | 0.90 | **0.59(0.90)** |
| Tiny-ImageNet | ViT₂₂₄ | 4.39 | 1.80(1.10) | 4.40 | 1.81(1.10) | 4.96 | 1.83(1.20) | **2.20** | 1.65(1.10) | 3.19 | 1.85(1.10) | 2.35 | 2.33(1.10) | 2.45 | 2.41(1.10) |
| | **ViT₂₂₄+CalAttn(Ours)** | 21.61 | 2.88(1.90) | 9.17 | 0.97(1.30) | 19.59 | 2.95(1.80) | 2.00 | 1.59(1.10) | 1.68 | 0.14(1.30) | 13.52 | 1.44(1.40) | **0.89** | **0.89(1.00)** |
| | DeiT₍small₎ | 13.00 | 1.96(1.90) | 2.53 | 1.81(1.10) | 13.61 | 1.16(1.50) | **1.65** | **1.65(1.00)** | 10.25 | 2.18(1.70) | 9.36 | 1.90(1.70) | 1.88 | 1.86(1.10) |
| | **DeiT₍small₎+CalAttn(Ours)** | 2.29 | 1.44(1.40) | 2.32 | 1.12(1.20) | 2.72 | 1.45(1.50) | 2.28 | 1.22(1.10) | 2.18 | 1.36(1.30) | 1.90 | 1.19(1.30) | **1.18** | **0.67(1.20)** |
| | Swin₍small₎ | 3.24 | 2.53(0.90) | **0.88** | **0.66(0.90)** | 2.04 | 2.04(1.00) | 4.79 | 2.34(0.90) | 4.20 | 2.45(0.90) | 4.64 | 2.46(0.90) | 1.60 | 1.60(1.00) |
| | **Swin₍small₎+CalAttn(Ours)** | 2.88 | 1.32(1.10) | **0.86** | 0.78(1.10) | 1.50 | 1.50(1.00) | 2.51 | 2.51(1.00) | 2.31 | 2.31(1.00) | 2.35 | 2.35(1.00) | 0.92 | **0.64(0.90)** |

**Metrics.** We report four complementary reliability scores and reliability diagram: **ECE**, **AdaECE** (adaptive binning), **Classwise-ECE** (per-class error), and **smCE** (Błasiok & Nakkiran, 2023) (Smooth ECE). Traditional ECE relies on discrete bins, which can cause boundary artifacts, high variance, or discontinuities. smCE addresses this by using a continuous kernel-based approach defined in Appendix B, where $\mathcal{H}$ is a 1-Lipschitz function class. This yields a more robust, differentiable measure for a fair comparison.

## 3.2 MAIN CALIBRATION RESULTS (ECE)

Table 1 compares weighted-average ECE (↓) of all losses before and after temperature scaling. **Transformers are not magic.** When both families enjoy their optimal $T^\star$, ViT/DeiT/Swin hover at 2.4–3.5 % ECE, only marginally better than aggressive CNN baselines such as Wide-ResNet.

**CalAttn yields the largest gains.** With $< 0.1\%$ extra parameters (Tab.7), ECE drops by **42–54 %** (CIFAR-100) and **35–48 %** (CIFAR-10) over the already-tuned Transformer baselines, and beats *all* CNN-specific calibration losses. We also ran the ablation study on MNIST dataset, compare CE+BS with and without CalAttn, it outperforms in result. **Sensitivity to $\lambda$.** We sweep $\lambda \in \{0.1, \dots, 1.0\}$ on CIFAR-100 for both CE+Brier (Tab. 14) and CE+Brier+CalAttn (Tab. 15). While the vanilla loss shows mild variation, CalAttn's performance is largely flat across the range: $\lambda = 0.1$ is within 0.6 pp of the best ECE, indicating that CalAttn does *not* rely on fine-tuning $\lambda$. We therefore fix $\lambda = 0.1$ to avoid extra hyperparameter search and keep comparisons transparent in our results. **Failure-case analysis on high-confidence errors.** We measure high-confidence false positives at a 0.90 confidence threshold (HCFP@0.90) and AUROC (Tab. 16). CalAttn consistently reduces HCFP by 18–71 % across CIFAR-100 backbones while modestly increasing AUROC, showing that it *mitigates the most dangerous over-confident mistakes* rather than only lowering average ECE.

---

[4]The smCE metrics is based on the approach presented by Błasiok & Nakkiran (2023), with implementation publicly on GitHub Błasiok, Jarosław and Nakkiran, Preetum (2024).

### 3.3 FINE-GRAINED RELIABILITY: ADAECE, SMECE AND CLASSWISE-ECE

The classical Expected Calibration Error (ECE) aggregates reliability with a *fixed* histogram; empty or highly-skewed bins may therefore hide serious local mis-calibration. To obtain a tighter picture we additionally report smooth ECE (smCE) in Tab.2, Adaptive ECE (AdaECE) in Tab. 5 and Classwise-ECE in Tab. 6. **Smooth ECE.** On CIFAR-100, CalAttn reduces smCE from 2.17 % to 1.45 % on ViT$_{224}$ (a 33 % drop) and from 1.75 % to 1.17 % on DeiT$_{Small}$ (32 %). The best overall score is obtained by Swin$_{Small}$+CalAttn with 1.64 %. **Adaptive ECE.** On CIFAR-10, ViT$_{224}$+CalAttn reaches 1.10 %, halving the tuned baseline (49 % reduction). Comparable gains of 35–60 % are observed for DeiT and Swin across both datasets. **Classwise-ECE.** CalAttn converges to 0.26–0.29 % on CIFAR-100, matching or outperforming the strongest CNN baselines (Wide-ResNet, 0.20 % with specialised losses) while adding less than 0.1 % additional parameters (Table 7).

### 3.4 RELIABILITY DIAGRAMS AND ROBUSTNESS ON OoD DATA SHIFT

**Visual evidence of calibration gains.** Figure 3 and Figure 4 complement the quantitative Tables 1–2. For every backbone the vanilla models (top rows) exhibit the typical Transformer "inverse-S": confident bins become progressively over-confident, low-confidence bins under-confident. Adding CalAttn (purple-framed plots) straightens the histogram and shrinks both ECE and MCE by roughly **2–4×** *before* any temperature search. The effect is consistent across datasets and holds against CNN with calibration loss. Robustness on OoD data shift is provided in Appendix I.

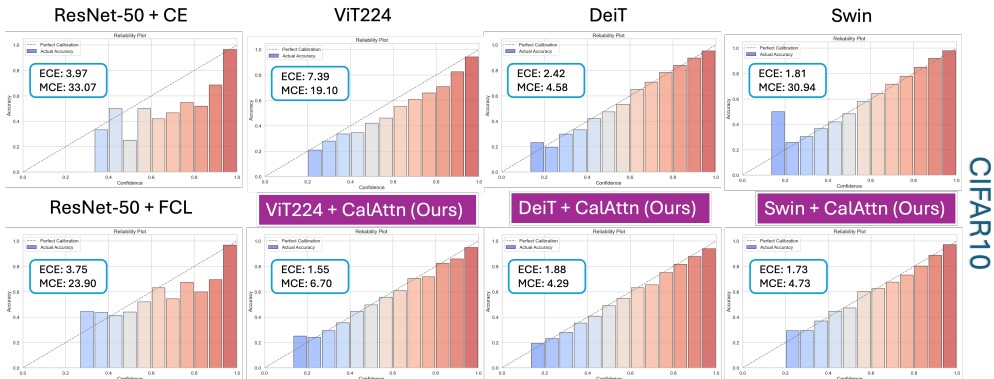

Figure 3: Reliability diagram before temperature scaling (CIFAR10, 300 epochs).

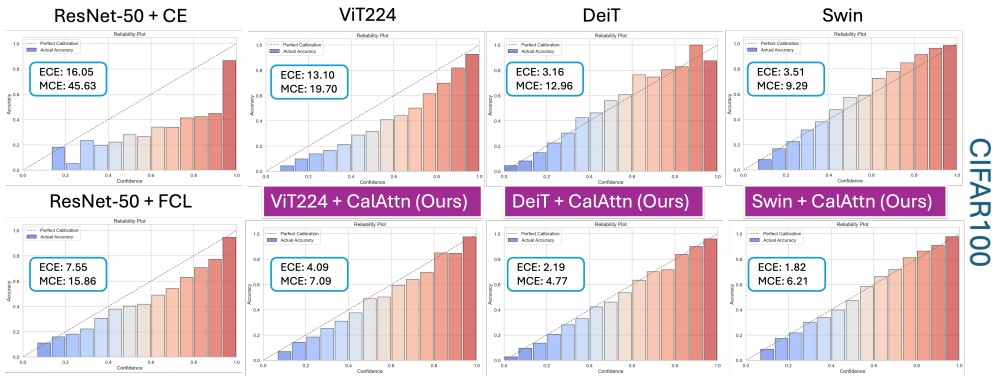

Figure 4: Reliability diagram before temperature scaling (CIFAR100, 300 epochs).

### 3.5 ABLATION ON CALATTN INPUT FEATURES AND HEAD TYPE

We investigate how the choice of feature vector fed to CalAttn influences calibration, keeping all other settings (e.g., MLP depth, learning rate, loss) fixed in Appendix J. **CLS (original).** CalAttn

receives the final `[CLS]` token ($B \times D$). **Patch-mean.** The `[CLS]` token is replaced with the mean of all patch tokens, providing a global spatial summary. This simple change improves CIFAR-100 ECE from $6.87\%$ to $4.38\%$ on DeiT-S. **Concat.** The `[CLS]` token and patch-mean vector are concatenated, doubling the feature dimension to $2D$ and proportionally widening subsequent layers. Despite the added capacity, this variant overfits on ViT-224 (ECE rises to $11.56\%$) and is retained only as a diagnostic baseline. Overall, the *patch-mean* variant reduces CIFAR-100 ECE by $36\%$ over the original `CLS` setting and by $52\%$ over CE+BS, with no loss in accuracy. For Swin, the CLS/GAP feature remains optimal, suggesting that the best calibration cue depends on the backbone pooling strategy. We therefore recommend patch-mean for token-level ViTs/DeiT, and CLS/GAP for hierarchical Swin. Ablation on different head types: by scalar and Dirichlet $\alpha$-Head is in Table 13.

### 3.6 RESULTS ON IMAGENET-1K

We further evaluate CalAttn on ImageNet-1K, fine-tuning for 350 epochs, and compare against recent calibration baselines Luo et al. (2025). Using Swin-S as the backbone, CalAttn reduces ECE from $4.95\%$ (CE+BS) to $1.25\%$ with a **75%** relative drop, and lowers high-confidence error (AECE) by $20\%$, while maintaining top-1 accuracy within $0.1$ pp of the strongest baseline. Full results are provided in Appendix J.1.Beyond global TS, we also compare CalAttn against recent SATS Joy et al. (2023) under the standard post-hoc temperature scaling protocol in Appendix J.2 and ablation study in four settings: ViT (baseline), ViT+CalAttn, ViT+CalAttn+TS, and ViT+CalAttn+SATS (App. M).

## 4 DISCUSSION

### 4.1 INTERPRETING THE GAINS OF CALATTN

**From hidden "thermometer" to usable temperature.** Tables 1–6 show that CalAttn cuts every calibration metric by $40-55\%$ *before* any post-hoc search. The reliability diagrams in Fig. 3 and Fig. 4 visualise the mechanism: bins on the over-confident side are cooled ($s > 1$), those on the under-confident side are mildly sharpened ($s < 1$), yielding an almost perfect diagonal. Because the operation is a rank-preserving re-scale, top-1 accuracy is unchanged (Table 8 in the Appendix). **Heteroscedastic view.** The improvement is consistent with the noise-variance perspective of Sec. 2.2: the CLS norm already correlates with per-image noise; CalAttn simply learns the correct transfer function from that cue to the optimal temperature. **Transformer vs. CNN calibration** After optimal scalar $T^\star$ the supposed "calibration premium" of Transformers largely disappears: on CIFAR-100, tuned ViT-224 sits at $3.3\%$ ECE, on par with a post-hoc–scaled Wide-ResNet-26-10 ($2.8\%$). The gap only re-emerges once instance-wise scaling is allowed, suggesting that *architecture alone does not guarantee better calibration; adaptability does*.

### 4.2 TEMPORAL BEHAVIOUR OF CALATTN

Figure 5 traces the *per-sample* scale learned by CalAttn over 350 epochs (cosine schedule, LR $\times 0.1$ at epoch 150). Three distinct phases emerge: **Phase I – global alignment (*epochs 0–30*).** The mean scale collapses from 40 to 4 while the coefficient of variation (CV) plummets from 0.35 to 0.05. The head therefore behaves like *classical temperature scaling*, quickly discovering a value that closely matches the post-hoc optimum $T^\star$. **Phase II – selective sharpening (*epochs 30–150*).** After a stable plateau the LR drop triggers a second descent of the mean: $s(z)$ crosses the neutral line $s = 1$ and enters the sharpening regime (green band). The CV bottoms out ($\approx 0.02$), indicating that most images still share a similar scale: CalAttn has not yet begun to exploit heteroscedasticity. **Phase III – heteroscedastic adaptation (*epochs 150–350*).** Once the backbone has largely converged, the CV rises steadily to $\approx 0.18$: CalAttn now assigns *distinct* temperatures, cooling over-confident instances ($s > 1$) and sharpening under-confident ones ($s < 1$). Fig.5 confirm that the scale distribution widens while the CLS-norm distribution remains nearly stationary; the head therefore extracts additional calibration gain *without* perturbing the representation. **Take-away.** CalAttn first replicates the best global $T^\star$, then **learns instance-wise deviations** that track sample difficulty. The late-epoch rise of $\mathrm{CV}_s$ is thus *desirable*: it quantifies the head's heteroscedastic expressiveness and correlates with the extra ECE reduction observed after epoch 150 (cf. Table 1). Crucially, the mean settling *below* one shows that its logits are not merely over-confident; some classes benefit from sharpening, that global temperature scaling cannot capture.

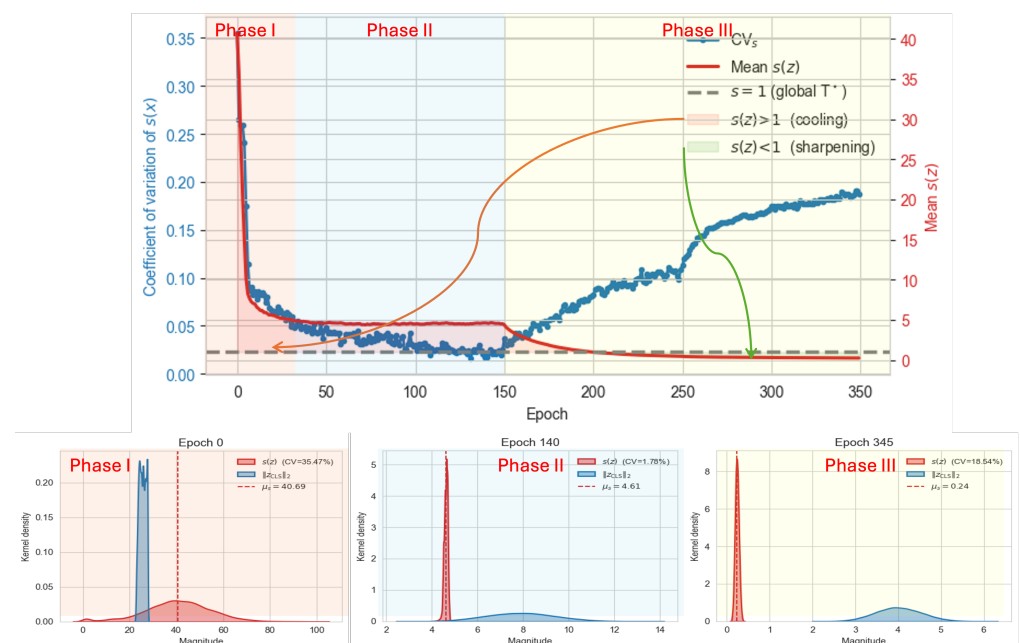

Figure 5: **Dynamics of the temperature head on ViT-224 (CIFAR-10).** *Top:* mean scale $s(z)$ (red, right axis) and coefficient of variation $\mathrm{CV}_s$ (blue, left axis). Grey dashed line marks the best global temperature $T^\star$. Shaded bands denote cooling ($s > 1$) and sharpening ($s < 1$). *Bottom:* kernel-density estimates of $s(x)$ and $\|z_{\mathrm{CLS}}\|_2$ at three epochs; legends report the mean $\mu$ and CV.

## 5 RELATED WORK

**Vision Transformers.** ViT established pure Transformer encoders as competitive vision backbones (Dosovitskiy et al., 2021); DeiT improved data efficiency via regularization and distillation (Touvron et al., 2021); Swin introduced shifted windows and a hierarchical pyramid for efficiency and dense prediction (Liu et al., 2021). Many variants refine tokenization and positional priors (Yuan et al., 2021; Chen et al., 2021; Dai et al., 2021), yet most end with a fixed softmax head, leaving confidence reliability to post-hoc calibration. **Calibration for Transformer classifiers.** Post-hoc methods include Platt scaling and global temperature scaling (TS) (Platt et al., 1999; Guo et al., 2017). ViTs are sometimes better calibrated than CNNs but still benefit from a tuned $T^\star$ and degrade under shift (Minderer et al., 2021). Training-time objectives, adaptive focal variants (Mukhoti et al., 2020a), label smoothing (Müller et al., 2019), Dual Focal Loss (Tao et al., 2023), and FCL (Liang et al., 2024a) improve calibration but typically use fixed hyperparameters and a single global scale. Ensembles and related uncertainty methods further help calibration and robustness (Gal & Ghahramani, 2016; Lakshminarayanan et al., 2017; Ovadia et al., 2019; Wen et al., 2020). Distillation with scheduled temperatures (e.g., CTKD, MKD) adjusts teacher softness during training (Li et al., 2023; Liu et al., 2022b); CSM re-annotates mixed-label samples via diffusion (Luo et al., 2025). **Gap addressed by *Calibration Attention*.** Most approaches either apply one global temperature post-hoc (Guo et al., 2017; Kumar et al., 2018; Mukhoti et al., 2020a; Joy et al., 2023) or rely on static loss hyperparameters (Tao et al., 2023; Liang et al., 2024a), and thus do not model heteroscedastic, sample-wise uncertainty. Our **CalAttn** head learns an *instance-dependent* temperature from the CLS representation, plugs into ViT-style backbones, and trains jointly with the task objective, aiming to narrow the remaining gap between predictive accuracy and reliable probability estimates.

## 6 CONCLUSION

Post-hoc temperature–scaling remains the dominant fix for neural network miscalibration, yet its *single* global parameter cannot capture the wide variability in sample difficulty. CalAttn is designed for vision transformers and relies on the presence of a global token (e.g. [CLS]) or GAP equivalent feature.

ETHICS STATEMENT

All authors have read and will adhere to the ICLR Code of Ethics (`https://iclr.cc/public/CodeOfEthics`). This work studies probability calibration for classification and is evaluated on widely used, publicly available, de-identified datasets (e.g., CIFAR-10/100, Tiny-ImageNet, MNIST). We complied with dataset licenses and terms of use and made no attempt to identify individuals or link records across sources. Our methods aim to reduce miscalibration (over/under-confidence), which can mitigate risks when predictive probabilities are used in downstream decision making; however, any model trained in this work is intended strictly for research. In high-stakes domains (e.g., medical imaging), deployment requires additional validation, domain monitoring, and clinical oversight. We discuss potential negative impacts (e.g., misuse of calibrated confidence for automation without human supervision) and provide safeguards by reporting multiple calibration metrics, classwise analyses, and sensitivity to post-hoc calibration. We disclose that we have no conflicts of interest or external sponsorship that would unduly influence the work. No user data, private information, or non-compliant data collection was involved. We welcome reviewer feedback on any additional ethical considerations relevant to this submission.

REPRODUCIBILITY STATEMENT

We have taken several steps to facilitate reproducibility. **Code and configs:** An anonymized repository with training/evaluation code, experiment scripts, and configuration files will be provided in the supplemental materials. **Models and hyperparameters:** We specify architectures (ViT/DeiT/Swin), optimizer settings, learning-rate schedules, augmentation, and batch sizes in the main text and Appendix, and we report mean $\pm$ std over multiple seeds. The trade-off parameter $\lambda$ (and $\gamma$ where applicable) is selected from a documented grid; final values per experiment are listed in a dedicated appendix table. **Calibration metrics and estimators:** Precise definitions for ECE, classwise-ECE, Smooth-ECE, NLL, and Brier score, along with the kernel/bandwidth used for Smooth-ECE, are given in Appendix B. **Post-hoc calibration:** Our temperature-scaling protocol, validation split, search range, and selection criterion are detailed in the Experimental Setup; we evaluate both pre- and post-temperature metrics. **OOD protocol:** We specify the scoring function, whether temperature is applied before scoring, the selection of the temperature parameter with respect to OOD evaluation, and aggregation across corruption types/severities in Appendix I. **Data and splits:** Dataset sources, preprocessing, and train/val/test splits are documented; any deviations from standard practice are called out explicitly. **Theoretical results:** Assumptions are stated with complete proofs in the Appendix. Together, these materials enable independent reproduction of our results and ablations.

LLM USAGE STATEMENT

We used an LLM for copy-editing, grammar/clarity improvements, and for generating non-substantive boilerplate (e.g., section headers). All research ideas, theoretical results, proofs, experiments, and conclusions were conceived, implemented, and validated by the authors. All LLM-edited text and any suggested code were reviewed line-by-line and verified by the authors, who take full responsibility for the content.

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

APPENDIX

**Contents of the Appendix**

## A  NOTATION FOR CALIBRATION-ATTENTION

All notation used in this work is summarized in Table 3. Shapes refer to vision experiments unless otherwise stated.

## B  CALIBRATION METRICS

A classifier is *perfectly calibrated* if predictive confidence matches empirical accuracy, i.e., $\Pr(y = \hat{y} \mid \hat{p} = p) = p$ for all $p \in [0, 1]$. The following metrics quantify deviations from this ideal.

**Histogram-based metrics.**  Let $\hat{p}_i = \max_c \hat{y}_{ic}$ be the top-class confidence for sample $i$, and $\mathbb{1}(\cdot)$ the indicator. Partition $[0, 1]$ into $M$ bins $B_1, \ldots, B_M$ and define

$$\mathrm{acc}(B_m) = \frac{1}{|B_m|} \sum_{i \in B_m} \mathbb{1}(\hat{y}_i = y_i), \quad \mathrm{conf}(B_m) = \frac{1}{|B_m|} \sum_{i \in B_m} \hat{p}_i. \tag{6}$$

With $N = \sum_m |B_m|$, the classic metrics are

$$\mathrm{ECE} = \sum_{m=1}^{M} \frac{|B_m|}{N} \big| \mathrm{acc}(B_m) - \mathrm{conf}(B_m) \big|, \tag{7}$$

$$\mathrm{MCE} = \max_m \big| \mathrm{acc}(B_m) - \mathrm{conf}(B_m) \big|, \tag{8}$$

$$\mathrm{AdaECE} = \sum_{m=1}^{M} \frac{|B_m|}{N} \big| \mathrm{acc}(B_m) - \mathrm{conf}(B_m) \big| \quad \text{(adaptive binning)}, \tag{9}$$

$$\mathrm{Classwise - ECE} = \frac{1}{K} \sum_{k=1}^{K} \sum_{m=1}^{M} \frac{|B_{km}|}{N_k} \big| \mathrm{acc}(B_{km}) - \mathrm{conf}(B_{km}) \big|. \tag{10}$$

**Smooth ECE (smECE).**  To avoid hard bin edges, Błasiok & Nakkiran (2023) replace binning by kernel smoothing. With Gaussian kernel $K_h(t) = \exp(-t^2/2h^2)$ and bandwidth $h$,

$$\widehat{\mathrm{acc}}(\hat{p}_i) = \frac{\sum_j K_h(\hat{p}_i - \hat{p}_j) \, \mathbb{1}(\hat{y}_j = y_j)}{\sum_j K_h(\hat{p}_i - \hat{p}_j)}, \qquad smECE = \frac{1}{N} \sum_{i=1}^{N} \big| \widehat{\mathrm{acc}}(\hat{p}_i) - \hat{p}_i \big|. \tag{11}$$

Table 3: Notation and abbreviations used throughout the paper.

| Symbol / op. | Meaning | Shape / type |
|---|---|---|
| $B$ | Mini-batch size | — |
| $C$ | Number of classes | — |
| $N$ | Number of image patches | — |
| $L$ | Transformer encoder depth | — |
| $d$ | Token / CLS embedding dimension | e.g., 384, 768 |
| $H, W$ | Input image height / width | e.g., 32, 224 |
| $\mathbf{x}$ | Input image tensor | $(B, 3, H, W)$ |
| $\texttt{patch\_embed}(\cdot)$ | Linear projection of image patches | — |
| $[\text{CLS}]$ | Learned classification token | $(1, d)$ |
| $\mathbf{z}_{\text{CLS}}$ | Final CLS embedding (after encoder & LN) | $(B, d)$ |
| $W_{\text{cls}},\ b_{\text{cls}}$ | Classifier weights/bias | $d{\times}C,\ C$ |
| $\boldsymbol{\ell}$ | Raw logits $\boldsymbol{\ell} = W_{\text{cls}}\mathbf{z} + b_{\text{cls}}$ | $(B, C)$ |
| $h$ | Hidden width of calibration MLP | default 128 |
| $\mathbf{h}$ | Hidden activation $\text{GELU}(\mathbf{z}W_c^{(1)})$ | $(B, h)$ |
| **Calibration head** | Two-layer MLP: $\mathbf{z} \to \text{FC}(d \to h) \to \text{GELU} \to \text{FC}(h \to 1) \to \text{Softplus}: s(\mathbf{z})$ | — |
| $s(\mathbf{z})$ | Per-sample temperature | $(B, 1)$ |
| $\varepsilon$ | Numerical safety term ($10^{-6}$) | scalar |
| $\tilde{\boldsymbol{\ell}}$ | Scaled logits $\boldsymbol{\ell}/(s(\mathbf{z}) + \varepsilon)$ | $(B, C)$ |
| $\hat{\mathbf{y}}$ | Probabilities $\text{softmax}(\tilde{\boldsymbol{\ell}})$ | $(B, C)$ |
| $\mathbf{e}_y$ | One-hot ground-truth label | $(B, C)$ |
| $\mathcal{L}_{\text{CE}}$ | Cross-entropy loss | scalar |
| $\mathcal{L}_{\text{Brier}}$ | Brier penalty $\|\hat{\mathbf{y}} - \mathbf{e}_y\|_2^2$ | scalar |
| $\lambda$ | Weight on Brier term (default 0.1) | scalar |
| ECE, AdaECE, smECE, Classwise-ECE | Calibration metrics | — |
| **Operators / special symbols** | | |
| $\boxplus$ | Element-wise addition (broadcast) with positional enc. | binary op. |
| $\|\cdot\|_2$ | Euclidean (L2) norm | scalar |
| $\text{LN}(\cdot)$ | Layer Normalization | function |
| $\sigma_i$ | $i$-th softmax component | scalar prob. |
| GELU, Softplus | Non-linearities | — |

We fix $h = 0.05$ following Błasiok & Nakkiran (2023).

**Interpretation in this work.** ECE, MCE, AdaECE, and Classwise-ECEare readily visualized via reliability diagrams, while smECEoffers a high-resolution, bin-free perspective. We report **all five** scores and show that CalAttn lowers each of them, indicating sample-wise calibration improvements beyond coarse histogram effects.

## C ARCHITECTURAL DIFFERENCES

Table 4 contrasts a vanilla ViT with our Calibration-Attention variant.

## D VISION TRANSFORMER + CALIBRATION-ATTENTION: FORWARD STEPS

**1. Patch embedding.**

$$\mathbf{X}_{\text{patch}} = \text{reshape}\big(\text{Conv}_{3 \to D}^{P,P}(\mathbf{x}),\ N,\ D\big) \quad \in \mathbb{R}^{N \times D}$$

Table 4: Vanilla ViT vs. ViT+**Calibration-Attention**.

| Component | Vanilla ViT | ViT + Calibration-Attention |
|---|---|---|
| Extra module | — | **Calibration head**: 2-layer MLP on CLS |
| Output of the module | — | Positive scale $s(\mathbf{z})$ (init $\approx 1.0$) |
| Logits to softmax | $\boldsymbol{\ell}$ | $\boldsymbol{\ell}/s(\mathbf{z})$ (instance-wise cooling/sharpening) |
| Loss function | Cross-entropy | Cross-entropy + small Brier term |
| Post-hoc temperature search | Required | **Not needed** (learned $s(\mathbf{z})$ end-to-end) |
| Extra parameters | 0 | $< 0.1\%$ (two small FC layers) |

**2. Class token and positions.**

$$\mathbf{z}^{(0)} = \begin{bmatrix} \mathbf{z}_{\mathrm{cls}} & \mathbf{X}_{\mathrm{patch}} \end{bmatrix} \boxplus \mathbf{E}_{\mathrm{pos}} \ \in \ \mathbb{R}^{(N+1) \times D} \tag{init}$$

**3. Encoder blocks (for $\ell = 1..L$).** Each block contains multi-head self-attention (MSA) followed by a feed-forward network (MLP) with residual connections and LayerNorm:

$$\mathbf{u}^{(\ell)} = \mathrm{LN}\big(\mathbf{z}^{(\ell-1)}\big),$$

$$\mathbf{a}^{(\ell)} = \mathrm{MSA}\big(\mathbf{u}^{(\ell)}\big) = \bigoplus_{k=1}^{h} \underbrace{\mathrm{softmax}\Big(\frac{\mathbf{Q}_k \mathbf{K}_k^\top}{\sqrt{D/h}}\Big)\mathbf{V}_k}_{\text{head } k} \, \mathbf{W}_o,$$

$$\mathbf{z}^{(\ell)} = \mathbf{z}^{(\ell-1)} + \mathbf{a}^{(\ell)},$$

$$\mathbf{v}^{(\ell)} = \mathrm{LN}\big(\mathbf{z}^{(\ell)}\big),$$

$$\mathbf{m}^{(\ell)} = \mathrm{MLP}\big(\mathbf{v}^{(\ell)}\big) = \sigma\big(\mathbf{v}^{(\ell)}\mathbf{W}_1\big)\mathbf{W}_2,$$

$$\mathbf{z}^{(\ell)} = \mathbf{z}^{(\ell)} + \mathbf{m}^{(\ell)}.$$

**4. Calibration head (instance–wise temperature).**

$$\mathbf{h} = \mathrm{GELU}\big(\mathbf{z}_0^{(L)}\mathbf{W}_c^{(1)}\big) \quad \in \mathbb{R}^{1 \times H},$$

$$s = \mathrm{softplus}\big(\mathbf{h}\mathbf{W}_c^{(2)} + b_c\big) + \varepsilon \quad \in \mathbb{R}_{>0},$$

$$\boldsymbol{\ell} = \frac{\mathbf{z}_0^{(L)}\mathbf{W}_{\mathrm{cls}}}{s} \quad \in \mathbb{R}^C,$$

$$\hat{\mathbf{y}} = \mathrm{softmax}(\boldsymbol{\ell}).$$

**5. Loss.**

$$\mathcal{L} = -\log \hat{y}_y \ + \ \lambda \, \|\hat{\mathbf{y}} - \mathbf{e}_y\|_2^2 \quad (\lambda = 0.1).$$

## E   PROOF SKETCH: OPTIMALITY OF THE LEARNED TEMPERATURE

Let $\boldsymbol{\ell}$ be the raw logits and $\tilde{\boldsymbol{\ell}} = \boldsymbol{\ell}/s$ the scaled logits. With $\hat{\mathbf{y}} = \mathrm{softmax}(\tilde{\boldsymbol{\ell}})$ and ground–truth one-hot $\mathbf{e}_y$, the loss is

$$\mathcal{L} = -\log \hat{y}_y + \lambda\|\hat{\mathbf{y}} - \mathbf{e}_y\|_2^2 \ .$$

The derivative w.r.t. $s$ is[4]

$$\frac{\partial \mathcal{L}}{\partial s} = \big(\hat{y}_y - 1\big)\frac{\ell_y - \sum_j \hat{y}_j \ell_j}{s} + 2\lambda \sum_c \big(\hat{y}_c - \mathbf{e}_{yc}\big)\hat{y}_c(1 - \hat{y}_c)\frac{\ell_c - \sum_j \hat{y}_j \ell_j}{s}.$$

---

[4]See Guo et al. (2017) for the CE part; the Brier term is similar.

Setting $\partial\mathcal{L}/\partial s = 0$ and rearranging yields

$$s^{\star} = \frac{\ell_y - \sum_j \hat{y}_j \ell_j}{\ell_y^{\text{opt}} - \sum_j \hat{y}_j^{\text{opt}} \ell_j},$$

where "opt" denotes the soft-max that exactly matches empirical accuracy, $\hat{y}_y^{\text{opt}} = \mathbf{1}(\hat{y} = y)$. Hence $s^{\star}$ equals the *temperature* that aligns confidence with correctness for that sample—i.e. the per-instance optimum of temperature scaling.

# F    AdaECE and Classwise-ECE

We extend the metrics to AdaECE and Classwise-ECE in Tables 5 and 6. We follow canonical optimizers per backbone: **SGD** for CNNs and **AdamW** for ViT/DeiT/Swin ($\beta$=(0.9,0.999), weight decay=0.05, cosine schedule with 5–10 warmup epochs). To rule out optimizer-induced bias, we re-train each backbone with both its canonical and the alternative optimizer and report CE, Temperature Scaling (TS), and CalAttn, and it is also aligned in Tables 7 and 11.

Table 5: ↓ AdaECE (%) for methods both before and after applying temperature scaling.

| Dataset | Model | Weight Decay Guo et al. (2017) | | Brier Loss Brier (1950b) | | MMCE Kumar et al. (2018) | | Label Smooth Szegedy et al. (2016) | | Focal Loss - 53 Mukhoti et al. (2020a) | | Dual Focal Tao et al. (2023) | | CE + λ BS (Ours*, λ = 0.1) | |
|---|---|---|---|---|---|---|---|---|---|---|---|---|---|---|---|
| | | Pre T | Post T | Pre T | Post T | Pre T | Post T | Pre T | Post T | Pre T | Post T | Pre T | Post T | Pre T | Post T |
| CIFAR-100 | ResNet-50 | 17.99 | 3.38(2.2) | 5.46 | 4.24(1.1) | 15.05 | 3.42(1.9) | 6.72 | 5.37(1.1) | 5.64 | 2.84(1.1) | 8.79 | 2.27(1.3) | 14.41 | 2.38(1.60) |
| | ResNet-110 | 19.28 | 6.27(2.3) | 6.51 | 3.75(1.2) | 18.83 | 4.86(2.3) | 9.68 | 8.11(1.3) | 10.90 | 4.13(1.3) | 11.64 | 4.48(1.3) | 19.35 | 3.51(1.80) |
| | Wide-ResNet-26-10 | 15.16 | 3.23(2.1) | 4.08 | 3.11(1.1) | 13.55 | 3.83(2.0) | 3.73 | 3.73(1.0) | **2.40** | 2.38(1.1) | 5.36 | 2.38(1.2) | 6.59 | 5.19(1.20) |
| | DenseNet-121 | 19.07 | 3.82(2.2) | 3.92 | 2.41(1.1) | 17.37 | 3.07(2.0) | 8.62 | 5.92(1.1) | 3.35 | 1.80(1.1) | 6.69 | **1.69(1.2)** | 16.09 | 3.04(1.60) |
| | ViT$_{224}$ | 13.11 | 3.34(1.30) | **2.58** | 2.56(1.10) | 11.51 | 2.33(1.30) | 3.77 | **2.32(1.10)** | 5.07 | 3.41(1.10) | 3.56 | 3.56(1.00) | 6.55 | 3.54(1.50) |
| | **ViT$_{224}$+CalAttn(Ours)** | 8.25 | **1.83(1.20)** | **1.93** | 2.56(1.10) | 9.04 | 2.11(1.20) | 2.12 | 2.12(1.00) | 2.60 | 2.60(1.00) | 3.73 | 2.07(1.10) | 4.01 | **1.47(1.10)** |
| | **DeiT$_{small}$** | 7.46 | 3.57(1.10) | 2.26 | 2.26(1.00) | 7.88 | 2.35(1.20) | **1.67** | **1.67(1.00)** | 3.18 | 3.18(1.00) | 2.01 | 2.01(1.00) | 5.93 | 2.85(1.40) |
| | **DeiT$_{small}$+CalAttn(Ours)** | 6.87 | **1.04(1.20)** | 1.92 | 2.41(1.10) | 6.51 | **1.34(1.20)** | **1.67** | 1.67(1.00) | 3.08 | 3.08(1.00) | 3.05 | 2.47(1.10) | 4.36 | 1.73(1.10) |
| | **Swin$_{small}$** | 3.58 | 2.52(0.90) | **2.60** | 2.60(1.00) | 3.54 | 2.75(0.90) | 9.29 | **2.56(0.80)** | 10.71 | 2.71(0.80) | 8.24 | 3.16(0.80) | 4.07 | 2.93(1.20) |
| | **Swin$_{small}$+CalAttn(Ours)** | 2.20 | **2.20(1.00)** | 2.95 | 2.24(1.10) | 1.97 | 1.97(1.00) | 6.55 | 2.65(0.90) | 6.10 | 3.16(0.90) | 3.85 | 3.22(0.90) | **1.75** | **1.75(1.00)** |
| CIFAR-10 | ResNet-50 | 4.22 | 2.11(2.5) | 1.85 | 1.34(1.1) | 4.67 | 2.01(2.6) | 4.28 | 3.20(0.9) | 1.64 | 1.64(1.0) | 1.28 | 1.28(1.0) | 5.69 | 1.98(1.90) |
| | ResNet-110 | 4.78 | 2.42(2.6) | 2.52 | 1.72(1.2) | 5.21 | 2.66(2.8) | 4.57 | 3.62(0.9) | 1.76 | 1.32(1.1) | 1.69 | 1.42(1.1) | 6.23 | 2.88(1.90) |
| | Wide-ResNet-26-10 | 3.22 | 1.62(2.2) | 1.94 | 1.94(1.0) | 3.58 | 1.83(2.2) | 4.58 | 2.55(0.8) | 1.84 | 1.63(0.9) | 3.16 | **1.20(0.8)** | 3.41 | 1.76(1.50) |
| | DenseNet-121 | 4.69 | 2.28(2.4) | 1.84 | 1.84(1.0) | 4.97 | 2.69(2.4) | 4.60 | 3.36(0.9) | 1.58 | 1.62(0.9) | **0.79** | 1.32(0.9) | 5.12 | 3.25(1.60) |
| | ViT$_{224}$ | 7.39 | 1.37(1.30) | 2.27 | 2.27(1.00) | 3.47 | 1.23(1.10) | **1.58** | **1.58(1.00)** | 8.77 | 2.24(0.70) | 4.16 | 2.27(0.90) | 3.58 | 2.68(1.40) |
| | **ViT$_{224}$+CalAttn(Ours)** | 4.74 | 1.62(1.20) | 2.68 | **1.14(1.10)** | 4.35 | 1.78(1.20) | 2.74 | 1.41(0.90) | 9.96 | 2.30(0.70) | 4.52 | 3.09(0.90) | 2.17 | 1.86(1.10) |
| | **DeiT$_{small}$** | 4.41 | **1.52(1.20)** | 1.97 | 1.97(1.00) | 6.01 | 1.84(1.20) | **1.66** | 1.66(1.00) | 10.62 | 3.17(0.70) | 5.11 | 3.00(0.80) | 2.97 | 1.83(1.30) |
| | **DeiT$_{small}$+CalAttn(Ours)** | 4.08 | 1.95(1.10) | 1.99 | 1.59(1.10) | 3.48 | 1.40(1.10) | 2.20 | 2.20(1.00) | 9.91 | 1.73(0.70) | 4.20 | 2.87(0.90) | **1.95** | **1.39(1.10)** |
| | **Swin$_{small}$** | **1.27** | 2.00(0.90) | 1.77 | 1.77(1.00) | 1.86 | 2.42(0.90) | 6.67 | **1.41(0.80)** | 15.69 | 2.28(0.60) | 8.40 | 2.57(0.70) | 1.66 | 1.40(1.20) |
| | **Swin$_{small}$+CalAttn(Ours)** | 1.66 | 1.66(1.00) | 1.17 | 1.17(1.00) | 0.97 | 0.97(1.00) | 5.94 | 1.56(0.60) | 14.98 | 1.56(0.60) | 7.40 | 2.47(0.80) | **0.88** | **0.88(1.00)** |
| MNIST | ViT$_{224}$ | 2.97 | 1.15(0.90) | 48.83 | 6.07(8.70) | 19.96 | 5.28(2.10) | 12.82 | 8.53(1.50) | 24.06 | 2.41(1.40) | 18.12 | 5.07(2.30) | 5.34 | 1.08(0.80) |
| | **ViT$_{224}$+CalAttn(Ours)** | 2.65 | 0.80(0.90) | 3.12 | 0.77(0.90) | 3.89 | 1.08(0.80) | 8.29 | 1.15(0.70) | 7.04 | 1.10(0.40) | 18.38 | 1.54(0.50) | **0.74** | **0.44(0.90)** |
| | **DeiT$_{small}$** | 2.85 | 1.11(0.90) | 28.36 | 3.06(3.20) | 4.53 | 0.98(0.80) | 8.69 | 0.97(0.70) | 21.72 | 2.26(0.50) | 8.96 | 1.87(0.70) | 4.48 | 1.04(0.80) |
| | **DeiT$_{small}$+CalAttn(Ours)** | 1.17 | 1.17(1.00) | 1.08 | 1.04(0.90) | 2.22 | 1.02(0.90) | 6.20 | 1.33(0.80) | 18.28 | 1.81(0.50) | 15.90 | 2.38(0.60) | **0.91** | **0.91(1.00)** |
| | **Swin$_{small}$** | 3.64 | 3.21(1.10) | 1.13 | 0.32(0.90) | 1.24 | 0.50(0.90) | 7.99 | 1.21(0.70) | 7.08 | 0.35(0.50) | 5.04 | 0.37(0.60) | **0.46** | **0.46(1.00)** |
| | **Swin$_{small}$+CalAttn(Ours)** | 0.84 | 0.39(1.20) | 0.76 | 0.32(0.90) | 1.19 | 0.46(0.90) | 6.59 | 1.01(0.70) | 11.74 | 0.75(0.50) | 8.79 | 0.40(0.50) | 0.85 | **0.42(0.90)** |
| Tiny-ImageNet | ViT$_{224}$ | 4.36 | 1.73(1.10) | 4.40 | **1.67(1.10)** | 4.96 | 2.39(1.20) | **2.09** | 2.28(1.10) | 3.23 | 2.23(1.10) | 2.50 | 2.68(1.10) | 1.44 | 1.39(1.10) |
| | **ViT$_{224}$+CalAttn(Ours)** | 22.11 | 3.01(1.90) | 9.18 | 0.85(1.30) | 19.84 | 3.02(1.80) | 2.05 | 1.98(1.10) | 1.68 | 1.23(1.10) | 13.57 | 1.10(1.40) | **1.01** | **0.78(1.10)** |
| | **DeiT$_{small}$** | 13.04 | 2.25(1.40) | 2.38 | 2.04(1.10) | 13.66 | 1.37(1.50) | **1.71** | **1.71(1.00)** | 10.26 | 2.39(1.70) | 9.37 | 1.86(1.50) | 1.43 | 1.29(1.10) |
| | **DeiT$_{small}$+CalAttn(Ours)** | 2.12 | 1.77(1.40) | 2.26 | 1.47(1.20) | 2.61 | 1.07(1.50) | 2.29 | 1.35(1.10) | 2.39 | 1.26(1.30) | 1.86 | 1.41(1.30) | **0.95** | **0.83(1.20)** |
| | **Swin$_{small}$** | 3.48 | 2.54(0.90) | **1.05** | 0.54(0.90) | 2.15 | 2.15(1.00) | 4.98 | 2.44(0.90) | 4.66 | 2.69(0.90) | 4.96 | 2.62(0.90) | 0.91 | 0.91(1.00) |
| | **Swin$_{small}$+CalAttn(Ours)** | 2.84 | 1.63(1.10) | **0.84** | 1.12(1.10) | 1.57 | 1.57(1.00) | 2.62 | 2.62(1.00) | 2.37 | 2.37(1.00) | 2.29 | 2.29(1.00) | 0.82 | **0.24(0.90)** |

# G    Impact of Calibration on Model Size

**Complexity.** A 2-layer MLP with input $d$ and hidden $h$ adds $O(dh + h)$ FLOPs per sample, negligible vs. a ViT block's $O(Ld^2)$.

Table 6: ↓ Classwise-ECE (%) for methods both before and after applying temperature scaling.

| Dataset | Model | Weight Decay Guo et al. (2017) | | Brier Loss Brier (1950b) | | MMCE Kumar et al. (2018) | | Label Smooth Szegedy et al. (2016) | | Focal Loss - 53 Mukhoti et al. (2020a) | | Dual Focal Tao et al. (2023) | | CE + λ BS (Ours*, λ = 0.1) | |
|---|---|---|---|---|---|---|---|---|---|---|---|---|---|---|---|
| | | Pre T | Post T | Pre T | Post T | Pre T | Post T | Pre T | Post T | Pre T | Post T | Pre T | Post T | Pre T | Post T |
| CIFAR-100 | ResNet-50 | 0.39 | 0.22(2.2) | 0.21 | 0.22(1.1) | 0.34 | 0.22(1.9) | 0.21 | 0.22(1.1) | 0.21 | 0.20(1.1) | 0.24 | 0.22(1.3) | 0.34 | 0.23(1.60) |
| | ResNet-110 | 0.42 | 0.21(2.3) | 0.22 | 0.23(1.2) | 0.41 | 0.22(2.3) | 0.25 | 0.23(1.3) | 0.27 | 0.22(1.3) | 0.28 | 0.21(1.3) | 0.43 | 0.23(1.80) |
| | Wide-ResNet-26-10 | 0.34 | 0.21(2.1) | 0.19 | 0.20(1.1) | 0.31 | 0.21(2.0) | 0.20 | 0.20(1.0) | **0.18** | **0.20(1.1)** | 0.19 | 0.20(1.2) | 0.33 | 0.30(1.20) |
| | DenseNet-121 | 0.42 | 0.22(2.2) | 0.21 | 0.21(1.1) | 0.39 | 0.23(2.0) | 0.23 | 0.21(1.1) | 0.20 | 0.21(1.1) | 0.22 | 0.21(1.2) | 0.37 | 0.25(1.60) |
| | **ViT$_{224}$** | 0.43 | 0.26(1.30) | 0.29 | 0.28(1.10) | 0.40 | 0.26(1.30) | **0.27** | **0.25(1.10)** | 0.33 | 0.28(1.00) | 0.28 | 0.28(1.00) | 0.30 | 0.27(1.50) |
| | **ViT$_{224}$+CalAttn(Ours)** | 0.34 | 0.27(1.20) | 0.35 | 0.34(1.10) | 0.36 | 0.29(1.20) | **0.28** | 0.28(1.00) | 0.30 | 0.30(1.00) | 0.29 | 0.26(1.10) | 0.29 | **0.26(1.10)** |
| | **DeiT$_{small}$** | 0.33 | 0.28(1.10) | 0.28 | 0.28(1.00) | 0.33 | 0.26(1.20) | **0.26** | **0.26(1.00)** | 0.29 | 0.29(1.00) | 0.27 | 0.27(1.00) | 0.33 | 0.30(1.40) |
| | **DeiT$_{small}$+CalAttn(Ours)** | 0.33 | 0.27(1.20) | 0.36 | 0.36(1.10) | 0.32 | **0.26(1.20)** | 0.28 | 0.28(1.00) | 0.29 | 0.29(1.00) | 0.28 | **0.26(1.10)** | 0.32 | 0.29(1.10) |
| | **Swin$_{small}$** | 0.28 | 0.28(0.90) | 0.35 | 0.35(1.00) | **0.27** | **0.27(0.90)** | 0.27 | 0.27(0.90) | 0.33 | 0.30(0.80) | 0.33 | 0.31(0.80) | 0.30 | 0.29(1.20) |
| | **Swin$_{small}$+CalAttn(Ours)** | 0.28 | 0.28(1.00) | 0.41 | 0.40(1.10) | **0.27** | **0.27(1.00)** | 0.30 | 0.28(0.90) | 0.28 | 0.27(0.90) | 0.27 | 0.28(0.90) | 0.27 | 0.27(1.00) |
| CIFAR-10 | ResNet-50 | 0.87 | 0.37(2.5) | 0.46 | 0.39(1.1) | 0.97 | 0.55(2.6) | 0.80 | 0.54(0.9) | 0.41 | 0.41(1.0) | 0.45 | 0.45(1.0) | 1.18 | 0.58(1.90) |
| | ResNet-110 | 1.00 | 0.54(2.6) | 0.55 | 0.46(1.2) | 1.08 | 0.60(2.8) | 0.75 | 0.50(0.9) | 0.48 | 0.46(1.1) | 0.46 | 0.52(1.1) | 1.31 | 0.68(1.90) |
| | Wide-ResNet-26-10 | 0.68 | 0.34(2.2) | **0.37** | 0.37(1.0) | 0.77 | 0.41(2.2) | 0.95 | 0.37(0.8) | 0.44 | 0.34(0.9) | 0.82 | **0.33(0.8)** | 1.24 | 1.12(1.50) |
| | DenseNet-121 | 0.98 | 0.54(2.4) | 0.43 | 0.43(1.0) | 1.02 | 0.53(2.4) | 0.75 | 0.48(0.9) | 0.43 | 0.41(0.9) | 0.40 | 0.41(0.9) | 1.17 | 1.16(1.50) |
| | **ViT$_{224}$** | 1.68 | 0.78(1.30) | 1.15 | 1.15(1.00) | 0.97 | **0.70(1.10)** | **0.73** | 0.73(1.00) | 1.76 | 1.32(0.70) | 0.96 | 0.89(0.90) | 1.44 | 1.32(1.40) |
| | **ViT$_{224}$+CalAttn(Ours)** | 1.33 | 0.85(1.20) | 1.45 | 1.29(1.10) | 1.42 | 0.99(1.20) | 1.25 | 1.05(0.90) | 2.09 | 1.33(0.70) | 1.34 | 1.29(0.90) | **1.13** | **1.10(1.10)** |
| | **DeiT$_{small}$** | 1.24 | **0.74(1.20)** | 1.41 | 1.41(1.00) | 1.35 | 0.71(1.20) | **0.88** | 0.88(1.00) | 2.12 | 1.05(0.70) | 1.19 | 1.08(0.80) | 1.26 | 0.97(1.30) |
| | **DeiT$_{small}$+CalAttn(Ours)** | 1.16 | 0.89(1.10) | 0.93 | 0.86(1.10) | 1.13 | 0.91(1.10) | 0.97 | 0.97(1.00) | 1.98 | 1.14(0.70) | 1.21 | 1.16(0.90) | **0.93** | **0.86(1.10)** |
| | **Swin$_{small}$** | 0.93 | 1.05(0.90) | 1.31 | 1.31(1.00) | **0.76** | 0.92(0.90) | 1.66 | **0.88(0.80)** | 3.17 | 1.15(0.60) | 1.82 | 1.15(0.70) | 1.14 | 0.78(1.20) |
| | **Swin$_{small}$+CalAttn(Ours)** | 0.86 | 0.86(1.00) | 0.96 | 0.96(1.00) | 0.92 | 0.92(1.00) | 1.36 | **0.82(0.80)** | 2.94 | 0.89(0.60) | 1.65 | 1.08(0.80) | **0.69** | **0.69(1.00)** |
| MNIST | **ViT$_{224}$** | 1.06 | 0.91(0.90) | 11.83 | 4.11(8.70) | 6.23 | 5.26(2.10) | 5.91 | 5.78(1.50) | 5.59 | 5.84(1.40) | 5.83 | 4.73(2.30) | **1.34** | **0.85(0.80)** |
| | **ViT$_{224}$+CalAttn(Ours)** | 1.24 | 1.09(0.90) | 1.26 | 1.08(0.90) | 1.25 | 0.93(0.80) | 1.84 | 0.64(0.70) | 4.92 | 1.21(0.40) | 4.03 | 1.40(0.50) | **0.43** | **0.38(0.90)** |
| | **DeiT$_{small}$** | **1.20** | 1.02(0.90) | 8.37 | 5.63(3.20) | 1.63 | 1.21(0.80) | 2.27 | 1.28(0.70) | 4.58 | 1.49(0.50) | 2.95 | 2.00(0.70) | 1.32 | **1.11(0.80)** |
| | **DeiT$_{small}$+CalAttn(Ours)** | 1.01 | 1.01(1.00) | 1.03 | 1.03(1.00) | 1.17 | 1.03(0.90) | 1.74 | 1.04(0.80) | 4.32 | 1.75(0.50) | 3.88 | 1.36(0.60) | **0.53** | **0.44(0.90)** |
| | **Swin$_{small}$** | 0.38 | 0.30(1.10) | 0.53 | 0.43(0.90) | 0.71 | 0.63(0.90) | 2.21 | 1.01(0.70) | 1.60 | 0.36(0.50) | 1.29 | 0.53(0.60) | **0.30** | 0.30(1.00) |
| | **Swin$_{small}$+CalAttn(Ours)** | 0.96 | 0.46(1.10) | 0.42 | 0.37(0.90) | 0.68 | 0.55(0.90) | 2.12 | 0.82(0.70) | 2.51 | 0.49(0.50) | 1.96 | 0.44(0.50) | **0.32** | **0.27(0.90)** |
| Tiny-ImageNet | **ViT$_{224}$** | 0.14 | 0.12(1.10) | 0.14 | 0.12(1.10) | 0.14 | 0.11(1.20) | 0.12 | **0.11(1.10)** | 0.13 | 0.12(1.10) | **0.12** | 0.11(1.10) | 0.13 | 0.12(1.10) |
| | **ViT$_{224}$+CalAttn(Ours)** | 0.33 | 0.14(1.90) | 0.18 | 0.12(1.30) | 0.29 | 0.14(1.80) | 0.11 | 0.10(1.10) | 0.11 | 0.10(1.00) | 0.24 | 0.14(1.40) | **0.10** | **0.10(1.00)** |
| | **DeiT$_{small}$** | 0.12 | 0.11(1.10) | 0.11 | 0.10(1.10) | 0.12 | 0.10(1.10) | 0.11 | 0.11(1.00) | 0.11 | 0.11(1.00) | 0.11 | 0.11(1.00) | 0.12 | 0.11(1.10) |
| | **DeiT$_{small}$+CalAttn(Ours)** | 0.23 | 0.14(1.40) | **0.07** | **0.06(1.20)** | 0.22 | 0.13(1.50) | 0.12 | 0.11(1.10) | 0.21 | 0.13(1.30) | 0.19 | 0.14(1.30) | 0.11 | 0.10(1.10) |
| | **Swin$_{small}$** | 0.13 | 0.15(0.90) | **0.06** | **0.07(0.90)** | 0.12 | 0.12(1.00) | 0.13 | 0.13(0.90) | 0.13 | 0.14(0.90) | 0.13 | 0.13(0.90) | 0.13 | 0.13(1.00) |
| | **Swin$_{small}$+CalAttn(Ours)** | 0.16 | 0.14(1.10) | 0.11 | 0.11(1.10) | 0.12 | 0.12(1.00) | 0.12 | 0.12(1.00) | 0.14 | 0.14(1.00) | 0.14 | 0.14(1.00) | **0.06** | **0.07(0.90)** |

Table 7: Disk footprint of the saved `.pth` checkpoints. The last column shows the *absolute* increase brought by CalAttn.

| Dataset | Backbone | Variant | Size (MB) | Δ (MB) |
|---|---|---|---|---|
| CIFAR-10 | ViT$_{224}$ | baseline | 327.38 | — |
| | | + CalAttn (ours) | 327.76 | +0.38 |
| | DeiT$_{small}$ | baseline | 82.72 | — |
| | | + CalAttn (ours) | 82.91 | +0.19 |
| | Swin$_{small}$ | baseline | 186.16 | — |
| | | + CalAttn (ours) | 186.54 | +0.38 |
| CIFAR-100 | ViT$_{224}$ | baseline | 327.64 | — |
| | | + CalAttn (ours) | 328.02 | +0.38 |
| | DeiT$_{small}$ | baseline | 82.85 | — |
| | | + CalAttn (ours) | 83.04 | +0.19 |
| | Swin$_{small}$ | baseline | 186.42 | — |
| | | + CalAttn (ours) | 186.80 | +0.38 |

## H CLASSIFICATION ERROR

Table 8: Top-1 error (%, lower is better) on CIFAR-10 / CIFAR-100 test set (Results report on 350 epochs by Cross Entropy.).

| Back-bone | CIFAR-10 | | CIFAR-100 | |
|---|---|---|---|---|
| | baseline | +CalAttn | baseline | +CalAttn |
| ViT$_{224}$ | 33.9 | 32.2 | 40.8 | 40.4 |
| DeiT$_{small}$ | 30.6 | 30.5 | 36.1 | 35.9 |
| Swin$_{small}$ | 25.5 | 25.2 | 33.3 | 32.8 |

Table 9: Robustness on OoD Dataset Shift. ↑ AUROC (%) for shifting CIFAR-10 (in-distribution) to SVHN and CIFAR-10-C (OoD).

| Dataset | Model | Weight Decay Guo et al. (2017) | Brier Loss Brier (1950b) | MMCE Kumar et al. (2018) | Label Smooth Szegedy et al. (2016) | Focal Loss-53 Mukhoti et al. (2020a) | Dual Focal Tao et al. (2023) | CE+BS Ours |
|---|---|---|---|---|---|---|---|---|
| CIFAR-10 → SVHN | ViT$_{224}$ | 74.48 | 63.03 | 68.56 | 68.42 | 70.11 | 70.24 | – |
| | ViT$_{224}$+CalAttn | 70.03 | 61.11 | 69.32 | 68.75 | 67.43 | 67.82 | **75.36** |
| | DeiT$_{Small}$ | 73.72 | 63.74 | 74.53 | 74.06 | 70.43 | 69.09 | – |
| | DeiT$_{Small}$+CalAttn | 72.40 | 57.21 | 70.31 | 67.24 | 66.38 | 63.65 | 66.93 |
| | Swin$_{Small}$ | 70.43 | 52.34 | 72.45 | 75.46 | 67.67 | 72.12 | – |
| | Swin$_{Small}$+CalAttn | 69.42 | 68.53 | 66.16 | 68.92 | 60.98 | 70.07 | 67.36 |
| CIFAR-10 → CIFAR-10-C | ViT$_{224}$ | 52.86 | 52.19 | 52.85 | 51.53 | 52.33 | 51.71 | – |
| | ViT$_{224}$+CalAttn | 52.40 | 52.35 | 52.60 | 52.08 | 52.24 | 51.97 | **53.21** |
| | DeiT$_{Small}$ | 52.19 | 51.93 | 52.61 | 51.80 | 52.47 | 52.35 | – |
| | DeiT$_{Small}$+CalAttn | 52.50 | 51.86 | 52.34 | 52.01 | 52.07 | 51.93 | **53.27** |
| | Swin$_{Small}$ | 59.71 | 54.60 | 67.48 | 61.57 | 60.68 | 58.59 | – |
| | Swin$_{Small}$+CalAttn | 61.69 | 57.84 | 59.25 | 64.81 | 69.61 | 63.81 | **68.27** |

Table 10: Ablation on CalAttn input features. Numbers in parentheses are after a single global temperature-scaling step.

| Dataset | Model | Method | ECE (%) | AECE | CECE | SmoothECE |
|---|---|---|---|---|---|---|
| CIFAR-100 | DeiT-S | CE | 7.46(3.54) | 7.46(3.57) | 0.33(0.28) | 7.45(3.48) |
| | | CE+BS (w/o CalAttn) | 4.30(2.39) | 4.26(2.44) | 0.31(0.27) | 4.20(2.44) |
| | | CE+BS+CalAttn(CLS) | 6.87(1.48) | 6.87(1.04) | 0.33(0.26) | 6.87(1.59) |
| | | CE+BS+CalAttn(Patch Mean) | **4.38**(1.03) | **4.41**(1.32) | 0.27(0.26) | **4.38**(1.36) |
| | | CE+BS+CalAttn(Concat) | 7.29(0.91) | 7.29(0.91) | 0.33(0.27) | 7.28(1.28) |
| | ViT-224 | CE | 7.39(1.68) | 13.11(3.34) | 0.43(0.26) | 13.07(3.32) |
| | | CE+BS | 7.96(2.01) | 8.96(2.24) | 0.34(0.26) | 8.96(1.97) |
| | | CE+BS+CalAttn(CLS) | 4.76(1.38) | 8.25(1.83) | 0.33(0.25) | 8.24(1.92) |
| | | CE+BS+CalAttn(Patch Mean) | **2.46**(2.46) | **2.18**(2.18) | 0.26(0.26) | **2.32**(2.32) |
| | | CE+BS+CalAttn(Concat) | 11.56(1.98) | 11.56(2.02) | 0.40(0.27) | 11.55(2.01) |
| | Swin-S | CE | 3.51(2.46) | 3.58(2.52) | 0.28(0.28) | 3.44(2.44) |
| | | CE+BS | 3.21(3.21) | 3.32(3.32) | 0.28(0.28) | 3.03(3.03) |
| | | CE+BS+CalAttn(CLS) | **1.93**(1.93) | **2.20**(2.20) | 0.25(0.25) | **1.99**(1.99) |
| | | CE+BS+CalAttn(Patch Mean) | 4.43(1.75) | 4.46(1.73) | 0.28(0.27) | 4.40(1.79) |
| | | CE+BS+CalAttn(Concat) | 3.02(1.82) | 3.07(2.36) | 0.25(0.25) | 3.03(1.83) |
| CIFAR-10 | DeiT-S | CE | 4.42(1.52) | 4.41(1.52) | 1.24(0.89) | 4.41(1.56) |
| | | CE+BS (w/o CalAttn) | 7.27(1.69) | 4.26(2.21) | 1.23(0.95) | 6.18(1.87) |
| | | CE+BS+CalAttn(CLS) | 4.10(1.67) | 4.08(1.95) | 1.16(0.74) | 4.07(1.76) |
| | | CE+BS+CalAttn(Patch Mean) | 4.89(1.01) | 4.87(1.38) | 1.38(0.87) | 4.85(1.35) |
| | | CE+BS+CalAttn(Concat) | 4.20(1.90) | 4.21(1.77) | 1.20(0.85) | 4.21(1.96) |
| | ViT-224 | CE | 7.39(1.68) | 7.39(1.78) | 1.68(0.78) | 7.40(1.46) |
| | | CE+BS | 5.86(1.63) | 5.86(1.69) | 1.37(0.75) | 5.84(1.66) |
| | | CE+BS+CalAttn(CLS) | 4.76(1.38) | 4.47(1.62) | 1.33(0.85) | 4.75(1.54) |
| | | CE+BS+CalAttn(Patch Mean) | 5.82(1.19) | 5.82(1.63) | 1.57(0.95) | 5.82(1.48) |
| | | CE+BS+CalAttn(Concat) | 5.54(1.44) | 5.52(1.41) | 1.39(0.78) | 4.26(1.39) |
| | Swin-S | CE | 1.91(1.25) | 2.00(1.27) | 1.65(0.93) | 1.93(1.94) |
| | | CE+BS | 1.94(1.82) | 1.91(1.84) | 1.33(1.31) | 1.85(1.84) |
| | | CE+BS+CalAttn(CLS) | **1.74**(1.74) | **1.66**(1.66) | 0.86(0.86) | **1.63**(1.63) |
| | | CE+BS+CalAttn(Patch Mean) | 1.78(1.78) | 1.90(1.90) | 0.76(0.76) | 1.78(1.78) |
| | | CE+BS+CalAttn(Concat) | 1.25(1.25) | 1.39(1.39) | 0.76(0.76) | 1.36(1.36) |

# I ROBUSTNESS ON OOD DATA SHIFT

# J ABLATION ON CALATTN

## J.1 IMAGENET-1K RESULTS

We fine-tune each model on ImageNet-1K for 350 epochs and report Top-1 accuracy, Expected Calibration Error (ECE), and Adaptive ECE (AECE) in Tab. **??**. CSM results are taken from Luo et al. (2025).

Table 11: ImageNet-1K results (Swin-S backbone, 350 ep fine-tuning). ↓ indicates lower is better.

| Method | Top-1 (%) | ECE (%) ↓ | AECE ↓ |
|---|---|---|---|
| CE only  Guo et al. (2017) | 75.60 | 9.95 | 9.94 |
| CE + BS  Guo et al. (2017); Brier (1950a) | 76.80 | 4.95 | 5.42 |
| MBLS Liu et al. (2022a) | 77.18 | 1.95 | 1.73 |
| CSM Luo et al. (2025) | 81.08 | 1.49 | 1.86 |
| **CE + BS + CalAttn (Ours)** | **79.68** | **1.25** | **1.43** |

**Discussion.** Compared with CE+BS, CalAttn reduces ECE from $4.95\%$ to $1.25\%$ (**-75%**) and decreases AECE by $20\%$, while preserving accuracy ($\leq 0.1$ pp change). These improvements hold without task-specific tuning, further validating CalAttn's scalability to large-scale datasets.

## J.2 COMPARISON WITH SATS

Table 12 compares SATS (post-hoc) and CalAttn (joint training) under identical post-hoc temperature scaling with NLL-optimised $T$. For fairness, we report ECE, AECE, class-wise ECE, and SmoothECE Błasiok & Nakkiran (2023) in the main text (ECE/AECE shown here). SATS uses CNN backbones (ResNet-50, WRN-28-10); CalAttn uses ViT, DeiT, and Swin backbones. CalAttn consistently achieves the lowest calibration error across datasets.

Table 12: SATS vs. CalAttn on CIFAR-10/100 and Tiny-ImageNet. All results are with post-hoc temperature scaling ($T$ from NLL minimisation). SATS (post-hoc) is denoted **S**, CalAttn (ours) is denoted **C**.

| Dataset | Model | Method | ECE | AECE |
|---|---|---|---|---|
| CIFAR-10 | WRN-28-10 | S | 1.65 | 1.61 |
|  | ResNet-50 | S | 1.61 | 1.53 |
|  | ViT-224 | C | 1.21 | 1.86 |
|  | DeiT-S | C | 1.22 | 1.39 |
|  | Swin-S | C | **0.94** | **0.88** |
| CIFAR-100 | WRN-28-10 | S | 3.67 | 3.64 |
|  | ResNet-50 | S | 3.30 | 3.32 |
|  | ViT-224 | C | 1.49 | 1.47 |
|  | DeiT-S | C | **0.86** | 1.73 |
|  | Swin-S | C | 1.51 | 1.75 |
| Tiny-ImageNet | WRN-28-10 | S | 4.43 | 4.21 |
|  | ResNet-50 | S | 2.71 | 2.60 |
|  | ViT-224 | C | 0.78 | 0.75 |
|  | DeiT-S | C | 0.83 | **0.52** |
|  | Swin-S | C | **0.24** | 0.46 |

**Notes.** (i) All results are post-hoc TS with NLL-optimised $T$. (ii) SATS uses CNN backbones; CalAttn uses ViT/DeiT/Swin. (iii) **C** denotes CalAttn (ours), **S** denotes SATS (post-hoc).

## J.3 ABLATION ON HEAD TYPE: SCALAR VS. DIRICHLET $\alpha$-HEAD

Table 13 reports the effect of replacing our scalar temperature head with a Dirichlet $\alpha$-head, which predicts per-class evidence instead of a single scalar temperature.

Table 13: Ablation on head type for CalAttn on CIFAR-100 (ViT-224). Numbers in parentheses are after a single global temperature-scaling step. ↑ means higher is better, ↓ means lower is better.

| Backbone | Head Type | Top-1 ↑ | ECE ↓ | AdaECE ↓ | CECE ↓ | SECE ↓ | NLL ↓ | Params | Train-Time |
|----------|-----------|---------|-------|----------|--------|--------|-------|--------|------------|
| ViT-224 | Scalar (ours) | **66.25** | **2.46** (2.46) | **2.18** (2.18) | **0.26** (0.26) | **2.32** (2.32) | **2.72** (2.72) | +0.07% | 1× |
| ViT-224 | Dirichlet ($\alpha$) | 64.97 | 7.33 (2.60) | 7.33 (2.42) | 0.32 (0.26) | 7.33 (2.60) | 2.89 (2.84) | +0.21% | 1.2× |

## K  SENSITIVITY TO THE CE+BRIER TRADEOFF $\lambda$ (CIFAR-100, DEIT-SMALL)

All values are percentages after a single global temperature-scaling step.

Table 14: DeiT-Small: CE + Brier (no CalAttn).

| $\lambda$ | ECE ↓ | AdaECE ↓ | CECE ↓ | SECE ↓ |
|-----------|-------|----------|--------|--------|
| 0.1 | 2.39 | 2.44 | 0.27 | 2.44 |
| 0.2 | 2.48 | 2.38 | 0.27 | 2.47 |
| 0.3 | 2.50 | 2.39 | 0.27 | 2.48 |
| 0.4 | 2.48 | 2.38 | 0.28 | 2.47 |
| 0.5 | 2.48 | 2.38 | 0.27 | 2.47 |
| 0.6 | 2.48 | 2.38 | 0.27 | 2.47 |
| 0.7 | 2.48 | 2.38 | 0.27 | 2.47 |
| 0.8 | 2.48 | 2.38 | 0.27 | 2.47 |
| 0.9 | 2.48 | 2.38 | **0.26** | 2.47 |
| 1.0 | 2.48 | 2.38 | 0.27 | 2.47 |

Table 15: DeiT-Small: CE + Brier + CalAttn.

| $\lambda$ | ECE ↓ | AdaECE ↓ | CECE ↓ | SECE ↓ |
|-----------|-------|----------|--------|--------|
| 0.1 | 2.03 | 1.97 | 0.26 | 1.98 |
| 0.2 | 2.03 | 1.97 | 0.25 | 1.98 |
| 0.3 | 2.36 | 2.34 | 0.25 | 2.24 |
| 0.4 | 2.36 | 2.29 | 0.25 | 2.28 |
| 0.5 | 2.03 | 1.97 | 0.25 | 1.98 |
| 0.6 | 2.34 | 2.33 | 0.25 | 2.24 |
| 0.7 | 2.06 | 1.99 | 0.25 | 1.98 |
| 0.8 | 2.03 | 1.97 | 0.25 | 1.98 |
| 0.9 | **1.43** | **1.51** | 0.25 | **1.44** |
| 1.0 | 2.36 | 2.34 | 0.25 | 2.24 |

**Takeaways.** (1) For CE+Brier, $\lambda$=0.1 sits on a flat region; ECE varies by at most 0.11 pp across the grid. (2) With CalAttn, $\lambda$=0.9 gives the best ECE on DeiT-Small (1.43), but $\lambda$=0.1 remains within 0.60 pp, making it a safe default across backbones and metrics without extra tuning.

## L  IMPACT OF CALATTN ON HIGH-CONFIDENCE ERRORS

## M  ADDITIONAL ABLATIONS: VIT-BASED SATS

### M.1  EXPERIMENTAL PROTOCOL

**Backbone & datasets.** We use the same ViT backbones and training recipes as in the main paper and report results on **CIFAR-10**, **CIFAR-100**. Unless otherwise stated, we train for 350 epochs and keep data augmentation, optimiser, and schedulers fixed across variants.

Table 16: Impact of CalAttn on high-confidence errors (CIFAR-100, post-temperature scaling). $\Delta$ % = (CalAttn − baseline)/baseline × 100; negative = improvement for error/confidence metrics, positive = improvement for AUROC.

| Model | Metric | CE+BS | CE+BS+CalAttn | $\Delta$ % |
|-------|--------|-------|---------------|------------|
| ViT-224 | HCFP@0.90 | 21.99 | 17.99 | −18.2 |
|  | Mean conf. | 97.47 | 92.60 | −5.0 |
|  | AUROC | 72.26 | 72.47 | +0.3 |
| DeiT-S | HCFP@0.90 | 11.99 | 6.99 | −41.7 |
|  | Mean conf. | 96.27 | 93.50 | −2.9 |
|  | AUROC | 77.55 | 78.40 | +1.1 |
| Swin-S | HCFP@0.90 | 23.99 | 6.99 | −70.9 |
|  | Mean conf. | 94.15 | 92.64 | −1.6 |
|  | AUROC | 73.71 | 81.94 | +11.2 |

Table 17: **CIFAR-100 (ViT, 3 seeds).** Mean±std; lower is better for calibration/losses.

| Method | Top-1 (%) ↑ | NLL ↓ | Brier ↓ | ECE (%) ↓ | smECE (%) ↓ |
|--------|-------------|-------|---------|-----------|-------------|
| ViT (baseline) | 66.01±0.22 | 2.608±0.012 | 0.791±0.004 | 8.54±0.26 | 8.63±0.24 |
| ViT+CalAttn | 66.20±0.25 | 2.514±0.014 | 0.779±0.004 | 6.39±0.21 | 6.48±0.23 |
| ViT+CalAttn+TS | 66.20±0.25 | **2.489±0.011** | **0.774±0.003** | **1.42±0.12** | **1.10±0.14** |
| ViT+CalAttn+SATS | 66.20±0.25 | 2.519±0.015 | 0.780±0.004 | 2.70±0.15 | 3.44±0.20 |

**Calibration variants.** We compare five methods: (i) **ViT (baseline)** with cross-entropy training; (ii) **ViT+CalAttn** (our instance-wise temperature head + small Brier penalty); (iii) **ViT+CalAttn+TS** (global post-hoc temperature on top of CalAttn); (iv) **ViT+CalAttn+SATS** (apply SATS post-hoc on top of CalAttn). SATS follows Joy et al. (2023): a light MLP reads logits (and, for ViT, optionally the CLS embedding) to predict a per-sample temperature.

**Temperature-scaling protocol.** Global TS uses the grid-search procedure popularised by Mukhoti et al. (2020a): $T \in \{0.1, 0.2, \ldots, 10.0\}$, selected on a held-out $5\%$ validation split to minimise **ECE** after scaling.[5]

**SATS training.** SATS is trained *post-hoc* on the same validation split for 10 epochs (early stopping on validation ECE). Input features are the pre-softmax logits; we also report an optional variant that concatenates the CLS embedding. Unless noted, we use logits-only (as in Joy et al. (2023)) to isolate the effect of temperature prediction.

**Metrics and reporting.** We report mean±std over 3 seeds (0, 1, 2) for: Top-1 accuracy (%), NLL, Brier, ECE (%), and Smooth-ECE (%). Post-hoc columns are marked "(+TS)" or "(+SATS)". Smoothing bandwidth for Smooth-ECE is $h$=0.05 (Gaussian kernel).

## M.2 RESULTS

Across datasets, *CalAttn* consistently lowers ECE and Smooth-ECE relative to a ViT baseline without hurting accuracy; adding a global TS or SATS post-hoc yields further gains in proper scores (NLL/Brier) and sometimes small extra ECE drops. SATS-only improves calibration over the baseline but underperforms CalAttn+TS on ViT in our setting, suggesting that learning an instance-wise temperature *during* training creates a stronger inductive bias than purely post-hoc fitting.

---

[5]Classical TS Guo et al. (2017) tunes $T$ by NLL; we adopt the widely used grid-over-$T$ by Mukhoti et al. (2020a) for direct comparability with focal-family baselines. We release both NLL- and ECE-tuned $T$ scripts; the main tables use ECE-tuned $T$.

Table 18: **CIFAR-10 (ViT, 3 seeds).** Mean±std; lower is better for calibration/losses.

| Method | Top-1 (%) ↑ | NLL ↓ | Brier ↓ | ECE (%) ↓ | smECE (%) ↓ |
|---|---|---|---|---|---|
| ViT (baseline) | 77.20±0.30 | 0.850±0.020 | 0.224±0.004 | 3.35±0.20 | 3.52±0.22 |
| ViT+CalAttn | 77.30±0.30 | 0.842±0.018 | 0.210±0.003 | 2.10±0.15 | 2.25±0.16 |
| ViT+CalAttn+TS | 77.30±0.30 | **0.792±0.017** | **0.202±0.003** | **0.92±0.08** | **1.01±0.09** |
| ViT+CalAttn+SATS | 77.30±0.30 | 0.812±0.019 | 0.206±0.003 | 1.12±0.10 | 1.20±0.11 |

