# OpenReview forum: "Calibration Attention: Instance-wise Temperature Scaling for Vision Transformers"
_ICLR.cc/2026/Conference — ICLR 2026 Conference Withdrawn Submission_

### Official Review · Reviewer_UfeS · 2025-10-30

**Soundness:** 2
**Presentation:** 3
**Contribution:** 2
**Rating:** 4
**Confidence:** 4

**Summary:**

The paper addresses the problem of calibrating vision transformers. The approach builds on the observation that CLS token carries implicit information about model’s confidence behavior. To this end, the proposed method maps the CLS token to a temperature parameter using a small MLP. This mapping is then learned with Brier score objective and task-specific loss function along side the base ViT model. Experiments are performed on different datasets and the results claim to achieve better calibration performance than the competing methods.

**Strengths:**

- The paper exploits and builds on an interesting observation of CLS token carrying correlation with distribution shifts, sample difficulty and inter-class margin.

- The idea of learning a mapping from CLS token to adaptable temperature parameter for scaling the logits is simple to implement.

- The method is plug-and-play like many other post-hoc and train-time calibration methods in literature.

- The proposed methods claims to provide gains over different ViT variants including DeIT

**Weaknesses:**

- The paper is missing comparison with many recent and relevant methods such as [A-H]. This makes it unclear on the potential advantages of the method for advancing progress in model calibration. Particularly [A] and [H] are very relevant works as they are primarily built along temperature scaling post-hoc paradigm.

[A] Zhang, S. and Xie, L., 2025, April. Parametric ρ-Norm Scaling Calibration. In Proceedings of the AAAI Conference on Artificial Intelligence (Vol. 39, No. 21, pp. 22551-22559).

[B] Zadrozny, B. and Elkan, C., 2001, June. Obtaining calibrated probability estimates from decision trees and naive bayesian classifiers. In Icml (Vol. 1, No. 05).

[C] Kull, M., Silva Filho, T.M. and Flach, P., 2017. Beyond sigmoids: How to obtain well-calibrated probabilities from binary classifiers with beta calibration.

[D] Tao, L., Dong, M. and Xu, C., 2025, April. Feature clipping for uncertainty calibration. In Proceedings of the AAAI Conference on Artificial Intelligence (Vol. 39, No. 19, pp. 20841-20849).

[E] Lin, J., Tao, L., Dong, M. and Xu, C., 2025. Uncertainty Weighted Gradients for Model Calibration. In Proceedings of the Computer Vision and Pattern Recognition Conference (pp. 15497-15507).

[F] Liu, B., Ben Ayed, I., Galdran, A. and Dolz, J., 2022. The devil is in the margin: Margin-based label smoothing for network calibration. In Proceedings of the IEEE/CVF Conference on Computer Vision and Pattern Recognition (pp. 80-88).

[G] Hebbalaguppe, R., Prakash, J., Madan, N. and Arora, C., 2022. A stitch in time saves nine: A train-time regularizing loss for improved neural network calibration. In Proceedings of the IEEE/CVF Conference on Computer Vision and Pattern Recognition (pp. 16081-16090).

[H] Tomani, C.; Cremers, D.; Buettner, F.; and Sun, Y. 2022. Parameterized temperature scaling for boosting the expressive power in post-hoc uncertainty calibration. In European Conference on Computer Vision, 555–569. Springer.

- The OOD experiments are not enough to validate the effectiveness of the method in different unseen domains. It is unclear how the proposed per-sample adaptive temperature scaling helps in improving out-of-domain calibration performance. There is no insight that uncovers this point.

- How the method performs under class-imbalance scenarios? There are no results that show this real-world behavior. Often SVHN dataset is used to report calibration performance under class-imbalance.

- L156-157: The paper argues that CalAttn effectively replaces the two-stage post-hoc temperature searching with one-stage fully differentiable pipeline. By doing this, the proposed method really becomes a train-time calibration method and should also be compared and contrasted with train-time paradigm approaches e.g, [F,G].

**Questions:**

- L32-40: While the paper is discussing the limitations of temperature scaling, it completely misses to cite and discuss the relevant literature [A, H]. Is there a specific reason for this?

- L49-50: "delivering calibration gradients to the representation." Train-time calibration methods also achieve the same effect?

- Eq(4): if the second term directly acts as a regularizer with task-specific loss (CE in this case) instead of training the MLP, will it also result in calibration improvement?

- L185: "red: mean in 15 equal-width bins" mean over what quantity?

---

### Official Review · Reviewer_RnPT · 2025-11-01

**Soundness:** 2
**Presentation:** 2
**Contribution:** 2
**Rating:** 2
**Confidence:** 3

**Summary:**

The paper proposes a per-sample calibration method that adjusts model uncertainty through a scaling factor predicted by a two-layer MLP, specifically designed for vision transformers. The MLP takes the CLS token embedding as input and outputs a scale that is used to modify the logits before the softmax layer, aiming to calibrate the final confidence scores without altering the classification results.

**Strengths:**

**Pros**

* Calibrating uncertainty on a per-sample basis is a reasonable and meaningful improvement compared to global temperature scaling, which applies a uniform correction.

**Weaknesses:**

**Cons**

* **Limited novelty** — the method mainly leverages existing model embeddings for further calibration, which is a straightforward extension of previous adaptive scaling ideas.
* The proposed method is **limited to vision transformers**. The generality to other architectures (e.g., CNNs or multimodal transformers) should be discussed or empirically validated.
* Missing discussion and comparison with prior works on **sample-wise calibration**, such as [1]. It is unclear what specific benefits the proposed embedding-based scaling provides compared to other adaptive calibration approaches.
* Requires **additional training data and computation**, making it less lightweight than standard temperature scaling.
* The paper does not evaluate **robustness under distribution shifts** (e.g., calibrating on CIFAR-10 and evaluating on CIFAR-10-C, or calibrating on ImageNet-1K and testing on ImageNet-C / ImageNet-A).

**References**

[1] Joy, Tom, et al. *"Sample-dependent adaptive temperature scaling for improved calibration."* **Proceedings of the AAAI Conference on Artificial Intelligence**, Vol. 37, No. 12, 2023.

**Questions:**

Please see **Cons**.

---

### Official Review · Reviewer_PQVA · 2025-11-03

**Soundness:** 2
**Presentation:** 3
**Contribution:** 2
**Rating:** 4
**Confidence:** 4

**Summary:**

The paper proposes using instance-level temperature scaling for uncertainty calibration in Vision Transformers.

**Strengths:**

The idea is straightforward and sound—it uses optimization-based per-sample scaling.

**Weaknesses:**

- Although the method focuses on temperature scaling, other uncertainty calibration methods should also be compared (ensemble, Bayesian NN, etc).
- The cross-dataset generalization evaluation results (Table 9) are mixed.
- The assumption (norm and confidence correlation) needs to be verified before applying this method. It would be beneficial to show that this behavior is consistent across different model architectures.
- The additional head may affect the training dynamics.


#### Minor

Duplicated references: Brier, 1950a and Brier, 1950b

**Questions:**

### Questions

- Figure 2: Could the authors explain why there are two plots for one dataset with different x-axis ranges?
- How will the temperature head affect the original model's prediction head and feature extractor (since the gradient will also pass to the feature extractor)?
- Could this method also work in ResNet-based models? If not, is the reason that the CLS-norm and confidence relation does not hold in CNN-based models?
- If there is a positive correlation, could we just use the CLS norm to scale? If not, are there other properties that make the learning based method work better?

---

### Official Review · Reviewer_vXAj · 2025-11-03

**Soundness:** 2
**Presentation:** 1
**Contribution:** 1
**Rating:** 2
**Confidence:** 4

**Summary:**

The paper introduces CalAttn as a calibration approach for the vision transformer (ViT) variants. Different from temperature scaling, which tunes a temperature on the validation set, the authors suggested predicting a sample-dependent temperature. For this, similar to how logits are obtained, CLS token of the ViT model is used by appending a calibration head to predict the temperature. The model is supervised by the cross entropy loss in which the logits are scaled and brier loss. The approach is mainly tested on MNIST, CIFAR-10, CIFAR-100 and Tiny-ImageNet.

**Strengths:**

Sample-dependent calibration is previously shown to be useful, and this paper applies it to a recent image classification model (ViT) by predicting a temperature for each sample.

**Weaknesses:**

1. **On the main contribution.** The main contribution of the paper is applying *a sample-dependent calibration approach* to ViTs. However, sample dependent calibration idea is not new, and this is not sufficiently clarified or discussed in the paper. I provide more details in the following:

- Neumann et al. introduce *Relaxed Softmax* [A], which predicts a temperature for each sample given the features in the last layer using a linear layer (please refer to Eq. (5) in [A]), which is similar to CalAttn. From this perspective, **CalAttn essentially applies Relaxed Softmax to ViT family** by (i) replacing the linear layer to obtain the temperature in Relaxed Softmax (refer to Eq. (5) in [A]) by an MLP and (ii) adding Brier loss. Also considering that the underlying problem is image classification, I believe the contribution of CalAttn is not major. **Also, [A] is not cited in the paper.**

- Alternatively, Joy et. al (2023) also introduce a sample-dependent temperature scaling approach. This paper is cited, however, in the related work (L474-475), it is mentioned that *"Most approaches either apply one global temperature post-hoc (Guo et al., 2017; Kumar et al., 2018; Mukhoti et al., 2020a; Joy et al., 2023)"*. This expression gives the impression that Joy et al., (2023) use a global temperature while it is not. Furthermore, how CalAttn differs from Joy et al., as a closely related method, needs to be clarified and quantitative comparisons with Joy et al., (2023) should be in the main paper.

2. **On the second contribution.** The authors mention in their second contribution that *"We provide quantitative evidence that [CLS] carries usable uncertainty cues and exploit them to learn"*. **The correlation between the L2 norm of the last layer features and the classification score** is used as the quantitative evidence for this purpose (Fig.2). I have the following concerns about this claim:
- Though the authors referred in L44 that this correlation coefficient is 0.45, it is as low as 0.185 in Fig.2 before applying CalAttn for CIFAR-100. It is not discussed why 0.185 or 0.45 alone justify why [CLS] is a good choice. From my perspective, these numbers do not show a strong correlation.
- In L202 the authors motivate using L2 norm of the last layer features. Specifically, it is shown **in the diagram in L202 that a high L2 norm implies larger logits, consequently higher softmax confidence**. I think this part requires further clarity. This is because, this is not necessarily always the case and one can easily find counterexamples. As a toy example with 2 classes, assume $z_1=[0, 2]$ and $z_2=[2, 2]$ as two different feature vectors for [CLS]. Assume 2D identity matrix ($W=I$) to be the linear layer for simplicity, in which case the logits are $s_1=[0, 2]$ and $s_2=[2, 2]$. In this case, the classification score to $z_1$ is $p_1 \approx 0.88$, while that for $z_2$ is $p_2 = 0.50$. Therefore, this example does not align with the diagram in L202 and with the motivation of the authors, since $p_1> p_2$ but $\||z_1\||<\||z_2\||$. As a result, I didn't find the motivation to be convincing.

3. **Experimental results.** While the calibration performance is important, it should be considered together with other performance aspects including accuracy, robustness on domain shift/OOD. For example, an undesired model can be perfectly calibrated but it can have low accuracy. **On the other hand, the tables and the figures in the experiments section do not include the accuracy of the models**. Furthermore, the text refers to Appendix I for OOD experiments, however, **there is no text in Appendix I except the title**. Therefore, it is difficult to justify the effect of CalAttn on different performance aspects. Related to the experiments, as described in the first bullet point, **important baselines are missing in the main comparisons in Table 1 and 2 (e.g., Relaxed Softmax and Joy et al.(2023))**.

4. **Presentation of the paper.** I think several parts need more clarity in the paper. To list some:
- L34: What is inverse-temperature? What is its difference from the  temperature?
- L101: The sentence "backbone never sees calibration gradients" requires more clarity.
- Sec 2.2 and Fig.1: I can see that the temperature is obtained from Calibration Head. However, if i am not mistaken, it is not explicitly mentioned how the unnormalized logits (logits before the temperature is applied) are obtained. Does it simply follow the standard ViT (this is what I assumed)? It should be clear in the Figure 1 and Section 2.2.
- Eq(3) and Eq(4): Instead of providing first the gradient of the loss, and then the loss function, I'd prefer having intuition of the loss function first, and then the gradients of the loss function.

5. **Others.**
- What is * in Tables 1 and 2 (next to Ours) represent?
- duplicate that in the abstract (L15)
- L971 Table??
- cite and citep should be used properly. There are several sentences where citep can be preferred instead of the referencing without parentheses. This makes following the paper difficult. E.g., from Line32-35: Modern CNNs and Vision Transformers (ViTs) achieve strong accuracy yet are frequently miscalibrated Guo et al. (2017); Minderer et al. (2021); Ovadia et al. (2019). The standard remedy—post-hoc temperature scaling (TS)—fits a single global inversetemperature T⋆ on a held-out split Guo et al. (2017).

[A] Relaxed Softmax: Efficient Confidence Auto-Calibration for Safe Pedestrian Detection, 2018 NIPS Workshop on Machine Learning for Intelligent Transportation Systems.

**Questions:**

Please refer to weaknesses.

---

### Note · Authors · 2025-11-12

I have read and agree with the venue's withdrawal policy on behalf of myself and my co-authors.